# Predator gaze captures both human and chimpanzee attention

Will Whitham[1,2°], Bradley Karstadt[3°], Nicola C. Anderson[4], Walter F. Bischof[4], Steven J. Schapiro[2], Alan Kingstone[4]*, Richard Coss[5], Elina Birmingham[3‡], Jessica L. Yorzinski[1‡]

1 Department of Ecology and Conservation Biology, Texas A&M University, College Station, Texas, United States of America, 2 Department of Comparative Medicine, UT MD Anderson Cancer Center, Bastrop, Texas, United States of America, 3 Faculty of Education, Simon Fraser University, Burnaby, British Columbia, Canada, 4 Department of Psychology, University of British Columbia, Vancouver, British Columbia, Canada, 5 Department of Psychology, University of California Davis, Davis, California, United States of America

☉ These authors contributed equally to this work.
‡ EB and JLY are joint senior authors on this work.
* alan.kingstone@ubc.ca

**Data Availability Statement:** All data files used in these analyses are available from the Open Science Framework database (DOI 10.17605/OSF.IO/WHSNE).

## Abstract

Primates can rapidly detect potential predators and modify their behavior based on the level of risk. The gaze direction of predators is one feature that primates can use to assess risk levels: recognition of a predator's direct stare indicates to prey that it has been detected and the level of risk is relatively high. Predation has likely shaped visual attention in primates to quickly assess the level of risk but we know little about the constellation of low-level (e.g., contrast, color) and higher-order (e.g., category membership, perceived threat) visual features that primates use to do so. We therefore presented human and chimpanzee (*Pan troglodytes*) participants with photographs of potential predators (lions) and prey (impala) while we recorded their overt attention with an eye-tracker. The gaze of the predators and prey was either directed or averted. We found that both humans and chimpanzees visually fixated the eyes of predators more than those of prey. In addition, they directed the most attention toward the eyes of directed (rather than averted) predators. Humans, but not chimpanzees, gazed at the eyes of the predators and prey more than other features. Importantly, low-level visual features of the predators and prey did not provide a good explanation of the observed gaze patterns.

## 1. Introduction

One of primates' most advanced visual skills is the rapid processing of human and nonhuman animals. Because high acuity color vision, like that of most primates, transduces far more visual information than can be actively attended to, mechanisms that restrict and bias attention to those stimuli of high survival relevance can be understood as an adaptation with positive fitness consequences (e.g., [1, 2]). Indeed, adult humans are more proficient at detecting other humans and animals relative to inanimate objects (e.g., cars, trees) in complex natural scenes

**Funding:** This work was supported by the National Science Foundation (BCS #1926327) to W.W., S. S., and J.Y., Natural Sciences and Engineering Research Council of Canada (RGPIN-2022-03079) to B.K., N.A., and A.K the funders had no role in study design, data collection and analysis, decision to publish, or preparation of the manuscript.

**Competing interests:** The authors have declared that no competing interests exist.

[1, 3, 4]. Likewise, human infants preferentially attend to animate items (e.g., police officers, lions), indicating that the prioritization of animacy emerges early on and is not learned [5]. It is even thought that this perceptual process is automatic in humans due to the tendency to look at animate objects even when they are irrelevant to the task [3, 6] and presented outside of conscious awareness [7].

These automatic attentional biases often involve stimuli of evolutionary consequence to primates. Human attention is often directed to threats, such as predators [8–10], hostile conspecifics [11], and dangerous objects [12]. Visual-search experiments have found that both human infants and adults are quicker to detect dangerous animals (e.g., snakes, spiders, lions) compared to non-dangerous animals [13–21]. Furthermore, [10] found that adults detected non-dangerous animals slower than dangerous animals because they spent more time looking at each of the dangerous (distractor) animals. Non-human primates exhibit attentional biases to animate stimuli, such as rhesus macaques' (*Macaca mulatta*) and Japanese macaques' (*Macaca fuscata*) vigilance to threatening versus non-threatening conspecific faces [22, 23], bonobos' (*Pan paniscus*) distraction due to task-irrelevant conspecific faces and leopards [24], and several primate species' biases to venomous animals like snakes (e.g., chimpanzees *Pan troglodytes*, gorillas *Gorilla gorilla gorilla*, and Japanese macaques: [25]; Japanese macaques: [26]; Japanese macaques: [27]). In many of these cases, the familiarity of the human participant or subject animal with any heterospecific animal stimuli was likely quite low (i.e., few encounters with real animals that would engender the attentional biases that were reported). These results have been taken to suggest that ancestral primates' early experience with predators modified the visual system to quickly detect dangerous animals [28, 29]. Threat detection can be influenced by experience, with human participants faster to detect modern threats (e.g., guns, syringes) compared to neutral stimuli even though these threats are too recent to have impacted our evolutionary history [12, 13].

Faces are often studied for their capacity to capture and hold attention compared to other body parts. Recognition of a face-like image is indubitably innate in humans as evidenced by preterm infants orienting more toward an inverted triangular array of lights than an upright triangular array of lights [30]. Human faces are detected quicker and more accurately by human participants than nonhuman animal faces in parallel visual search tasks [31, 32]. When presented with images of human faces, people tend to fixate (look at) the internal features, with a particular focus on the eyes (e.g., [33–37]). In a similar vein, adult humans, especially those with high anxiety, are slower to disengage their attention when viewing angry faces compared to happy or neutral ones [38, 39]. This preference to fixate the eyes appears early on, often within the first fixation [40, 41]. Like humans, chimpanzees also show an initial bias to fixate the eyes of conspecifics, but spend less time looking at the eyes compared to other parts of the conspecific face and compared to humans' gaze to eye regions [42, 43]. Bonobos, another great ape species, attend to conspecific eye regions to an even greater degree than do chimpanzees [44]. Humans and other primates may look at the eyes of others because eyes are powerful conveyors of social information, such as emotional and attentional states [45–47]. As a result gaze direction may also influence attention. Previous studies indicate that a staring gaze target was detected quicker among averted gaze distracters [48, 49], but see [50].

Proponents of a general face bias argue that it is essential for humans to prioritize animacy and faces alike given that ancestral humans had to be vigilant to both predators and other dangerous species as well as attentive to non-dangerous species, such as prey and domestic animals [1, 4, 31]. Although experimental research on the visual perception of animal faces in natural environments is limited, a handful of studies help to shed light on this area. For example, visibility of the head significantly improves the speed with which human participants detect animals in a visual search task, likely due to the expectation of a face [51]. Similarly, [52] found

that forward-facing animals (i.e., animals facing towards the viewer) were more rapidly detected by adult humans than animals facing away and that forward-facing predators were more rapidly detected than forward-facing prey. These findings suggest that, as with human faces, the bias to attend to nonhuman animal faces is especially strong when direction of gaze is aimed toward the observer. Research has shown that most faces, and face-like configurations like those of pareidolia images, generally appear to capture attention in humans and other primates [53–55]. Such a bias may be understood as part of a larger system of biases to predators and prey, and conspecifics and heterospecifics, and eyes and other features; all related to discerning the relative agency and intention that motivates the behaviors of the organism to which attention is directed [56]. Only one investigation [57] provides direct experimental evidence of the shared bias to fixate eyes of humans and non-human animals alike. However, the animal images used by Yarbus were unnatural (e.g., a penciled sketch of a lion's face, a photo of a gorilla sculpture) and it is therefore difficult to generalize the finding.

Many human studies present faces unnaturally ("passport-style") with body features absent below the neck, precluding contextual interpretation. When presented in this manner, the eyes are a visually conspicuous part of the image with their highly contrasting colors relative to the rest of the face [58]. As a result, participants may look toward the eyes automatically because of low-level visually salient features, rather than looking toward the eyes because of high-level social meaning or scene comprehension [59–61]. To test this hypothesis, [40] presented humans with complex real-world scenes containing one to three people and compared participants' initial fixations against the salience model using the Salience Toolbox [62]. The salience model operates by creating and combining several topographic feature maps (e.g., changes in intensity, color, and edge orientation) into one final salience map coding for conspicuous (i.e., salient) scene locations. [40] found that, even though participants' initial fixations frequently landed on the head and eyes, salience at the head and eye regions were lower than what would be expected by chance. Using the same salience model, [51] demonstrated that pictures of animals in complex natural environments also have comparatively low salience. Thus, bottom-up processing does not appear to accurately describe eye movements in complex real-world scenes containing humans or nonhuman animals.

The present study was an initial exploration of how both humans (Experiment 1) and chimpanzees (*Pan troglodytes*; Experiment 2) attend to images of a species with prototypical predator features (lions) and images of a species with more prototypical prey features (impala). We did not make specific predictions about any divergent patterns of gaze behavior for humans and chimpanzees, and the two experiments are successive attempts with different species to identify the attentional biases reviewed above. Experiment 2 was not motivated by extant primate behavioral ecology *per se.* Mature male lions (images of which acted as the prototypical predators for these experiments) have little hunting responsibility, share little overlap with extant chimpanzee populations in either geography or habitat, and are not known to regularly interact with chimpanzees in any natural context. Likewise, chimpanzees do not hunt impala. Instead, Experiment 2 was designed as an assay of primate attentional biases to the prototypical features described above and to the social signaling of directed and averted eyes. We anticipated a degree of attention to lion images from the chimpanzees in this study, since savanna-living chimpanzees in Tanzania treat lions as a predation threat and alarm call vigorously when they see or hear them [63], but our design was not structured to assess this particular bias (e.g., of lion features versus those of other big cats). Note also that impala images, while having characteristics of many prey features, are also not necessarily "neutral" images given the entwined evolutionary histories of ancestral primates and impala. As such the impala images function in these experiments as images without characteristic predator features, and

without features that should engender any conserved attentional bias related to threatening stimuli, but not as images without any evolutionary consequence.

We tested four predictions for anticipated patterns of gaze behavior of humans and chimpanzees to images of lions and impala. Three of our predictions were related to the anticipated response of humans and chimpanzees to the prototypically predatory features of the lion (e.g., forward-facing eyes, robust body) and the prototypically preyed-upon features of the impala (e.g., laterally-placed eyes). First, we predicted that participants would look at predators more than prey. Second, we predicted that the eyes of predators and prey would be the most fixated region in a natural image. Third, we predicted that fixations to the eyes will be influenced by both animal type (predator or prey) and the gaze direction of the animal. Given the importance of eyes for signifying threat for a variety of primates, including chimpanzees and humans [64–67], we expected that eyes of forward-facing predators will receive more fixations than the eyes of averted-facing predators, whereas fixations to the eyes of prey will not increase with direct gaze.

Our final prediction is related to the predictive power of visual brightness, contrast, color, and other low-level stimulus features (hereafter: salience). For our previous predictions, we assumed that any differences in patterns of gaze behavior among the predator and prey images and among the different regions of interest are due to differences in the way that the human or chimpanzee represents the content of the image. That is, we expect that gaze behaviors are caused by something about the lion being a lion, and the impala being an impala, and the orientation of the eyes. However, a more conservative hypothesis for research of this kind is always that any differences in gaze behavior are due to differences in visual salience. Effectively testing our first three predictions required testing whether our pattern of results could instead be understood as a consequence of, for example, some uniquely salient visual feature of a lion's physiognomy, like the color and contrast introduced by a mane. We used the Salience Toolbox [62] to compute salience maps based on the low-level visual features of each image, and to test our prediction that human and chimpanzee gaze behaviors to our stimuli would not be predicted by salience information in the images alone.

To test these predictions, we presented humans (Experiment 1) and chimpanzees (Experiment 2) with 96 complex natural scenes that contained either one female impala (a prototypical prey) or one male lion (a prototypical predator). The animals in these scenes were all fully visible, doing nothing except standing in an upright posture and forward-facing (i.e., toward the observer) or averted-facing (i.e., away from the observer). Gaze behaviors were recorded as humans and chimpanzees viewed the animal images.

## 2. Experiment 1

### 2.1 Materials and methods

**2.1.1 Participants.** Thirty-three undergraduate students from the University of British Columbia ranging in age between 19 and 27 years ($M = 19.61$, $SD = 1.66$; 28 females) participated in this study between January and March 2020. Ethnicity was reported as 55% Chinese, 9% Caucasian, 9% Indian, 9% East Asian, 6% Arab, 6% Korean, 3% Mexican, and 3% African. All had normal or corrected-to-normal vision and were naïve to the purpose of the experiment. Each participant received course credit for participation, and written, informed consent was acquired from every participant prior to their participation in the study. This experiment was approved by the University of British Columbia's Institutional Review Board (#H10-00527).

**2.1.2 Apparatus.** Eye movements were recorded at 1000Hz using a desktop mounted Eye-Link 1000 Plus tracking system (SR Research, Canada). The online saccade detector of the eye

tracker was set to detect saccades with an amplitude of at least 0.1˚, using an acceleration threshold of 8000˚/s$^2$ and a velocity threshold of 30˚/s. Average accuracy of the tracker is typically within 0.25˚ and 0.5˚.

**2.1.3 Procedure.** All participants completed two tasks: 'Free-view task' and 'Danger Rating task'. In the Free-view task, participants were instructed to simply look at the stimuli while being eye tracked. In the Danger Rating task, participants were instructed to look at and rate (see below) how dangerous the animal appeared, while being eye-tracked. Task order was counterbalanced across participants. Each block of both the Free-view task and Danger Rating task presented 96 images (24 lions with direct gaze, 24 lions with averted gaze, 24 impalas with direct gaze, 24 impalas with averted gaze) and lasted approximately 10 minutes. Images were randomised within each task without replacement. The eye tracker was recalibrated between blocks.

Participants sat in a brightly lit room and were placed in a chinrest so that they sat approximately 50 cm from the display computer screen. Before the experiment, a 9-point eye movement calibration procedure was conducted. Participants were instructed to fixate a central black dot, and to then fixate it again when it appeared at each of nine different locations on the screen. This calibration was then validated, a procedure that calculates the difference between the calibrated gaze position and target position and corrects for this error in future gaze position computations. After successful calibration and validation, the trials began.

At the beginning of each trial, a fixation point was displayed in the centre of the computer screen to correct for drift in gaze position. Participants were instructed to fixate this point and then press the spacebar to start a trial. One of 96 photographs was then shown in the centre of the screen and remained visible until 8 sec had lapsed. Note that this meant that when an image first appeared, the initial central fixation (which was discarded from analysis) landed approximately on the centre of the body of an animal. In the Free-view task, participants received no further instructions, and the image display was replaced with the drift correction screen after the 8 s. In the Danger Rating task, participants were asked to use a keyboard number pad to rate how dangerous the animal appeared (on an interval scale of 1–5 using the full range of the scale, with 1 being "Not dangerous at all" and 5 being "Extremely dangerous"). After indicating their response, the image display was then replaced with the drift correction screen. This process repeated until all images had been viewed. The participants therefore viewed each photograph twice (once in the Free-view task and once in the Danger Rating task). We found no main effects or interaction effects for task nor task order ($p > .05$), so we pooled the data from these two tasks to create a single data set. Because no significant differences in gaze behavior were observed between Danger Rating and Free View conditions, we did not incorporate the ratings into additional analyses. Danger Rating scores were higher for lions than impala, and higher for directed animals than averted ($ps < .05$).

**2.1.4 Stimuli.** Color digital photographs depicting lions and impala were obtained from online sources (Fig 1A). Each of the 96 photographs displayed either a male lion or a female impala. Half of the images contained animals with direct gaze and half with averted gaze. The direct-gaze images displayed animals with their eyes and heads oriented toward the camera; the averted-gaze images displayed animals with their eyes and heads looking to the side. The bodies of the animals were orientated with their sides partly or completely visible; due to limitations in available photographs, it was not possible to standardize the exact body orientation. The image backgrounds included plants, rocks, ground, or sky. In addition, in half of the averted-gaze images, the animals were facing to the left and in the other half, to the right. Regions of Interest (ROIs) were created in DataViewer (SR Research, Canada) using rectangles for the eye regions and freehand polygons for the head and body regions (Fig 1B). Photographs filled the entire screen (1024 x 768 pixels) of the eye-tracker monitor.

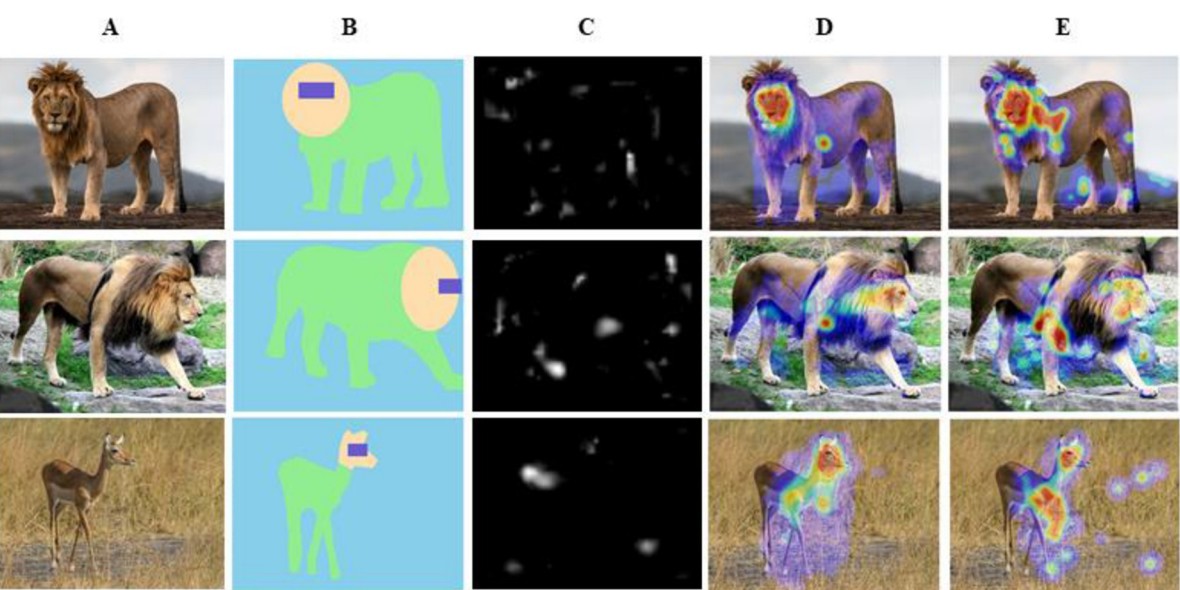

**Fig 1. Visualizations of experimental stimuli.** (A) Examples of three stimuli. From top to bottom: Lion with direct gaze, lion with averted gaze, and impala with averted gaze. (B) Regions of interest used in the analyses (eyes, head, body, background). (C) Salience maps [62]. Heatmaps (aggregated across all subjects) for gaze patterns of (D) humans and (E) chimpanzees. An image of an impala with averted gaze is not pictured due to photograph copyright issues. Lion with direct gaze printed under a CC BY license with permission from copyright holder Kanwar Deep Juneja. Lion with averted gaze printed under a CC BY license with permission from copyright holder Cass Womack. Impala with averted gaze printed under a CC BY license with permission from copyright holder Tris Enticknap.

**2.1.5 Eye-tracking analysis.** We defined the following ROIs: eyes (the portion of the head around, and including, the eyes of the animal), heads (the remaining portion of the head of the animal), body (including torso and legs but excluding eyes and head), and background (everything else, see Fig 1B). These ROIs were all within the margin of error for tracker accuracy. To determine which ROIs were of most interest to participants, we computed area-normalized fixation proportions by first dividing the number of fixations for a region by the region's area to create a *normalized fixation count value*; and then created a *normalized fixation proportion* value for each region by dividing the normalized count value for that given region by the sum of all the normalized count values across all regions [33, 34]. Use of area-normalized fixation proportions controls for potential confounds such as absolute ROI size differences or differences in the ratio of the size of one ROI to another within and across conditions, and have the advantage of incorporating gaze behaviors and ROI size into one dependent measure for further analyses.

To determine how participants first observed the visual scene, we examined the initial fixation made by participants, defined as the first fixation after making a saccade from the central fixation dot (i.e., fixation #2). Initial fixation proportions were calculated as the number of initial fixations in each region for one trial type (e.g., impala averted) divided by the total number of initial fixations for all regions in that same trial type. Initial fixation proportions were not area-normalized.

**2.1.6 Salience analysis.** The Salience Toolbox developed by [62] enabled the measurement of the visual salience of an image via strong changes in intensity, color and local orientation. To remain as consistent as possible with others using this toolbox, we used all default parameters. The final salience maps were scaled to the same size as the original fixation analysis (1024 x 768 pixels) using bilinear interpolation. Examples of scenes, their regions, and corresponding salience maps are shown in Fig 1. Salience values were normalized to a range of 0 (not salient)

to 1 (very salient). As visual salience has been hypothesized to have its greatest impact on the first saccade [68, 69], we focused our analysis on the first fixation made by participants after making a saccade from drift correction cross. We computed the average salience of fixated scene locations and compared this value to two control values. The first control value was the average salience of random locations sampled uniformly from the image (called "uniform-random"). To control for the known bias to fixate the lower central regions of scenes [70], the second control value was the average salience of random locations sampled from the smoothed probability distribution of all first-fixation locations from participants' eye movement data across all scenes (called "biased-random"). These comparisons allowed us to determine whether the salience model accounted for first fixation position above what would be expected by chance.

## 2.2 Experiment 1 results

**2.2.1 Statistical analysis.** First, to determine if participants displayed a general bias for fixating predators versus backgrounds, we aggregated normalized fixation proportions by region into a singular interest area for the whole animal (Section 2.2.2 Animals versus backgrounds). We performed a 2 (stimulus species: lion vs. impala) x 2 (region: animal or background) within-subjects analysis of variance (ANOVA). Second, to determine whether participants directed their gaze toward specific regions of the animals, we submitted all the normalized fixation proportion data to a 2 (stimulus species: lion vs. impala) x 2 (gaze direction: directed vs. averted) x 4 (ROI: eyes, head, body, background) within-subjects ANOVA (Section 2.2.3 Specific animal ROIs). Third, to assess whether participants directed their initial gaze toward specific regions of the animals, we performed another 2 (stimulus species: lion vs. impala) x 2 (gaze direction: directed vs. averted) x 4 (ROI: eyes, head, body, background) within-subjects ANOVA on the initial fixation proportion data (Section 2.2.4 Initial fixations on animals). We performed a comprehensive series of *a priori contrasts* to test our predictions–of differences between directed and averted gaze within and across each ROI and within and across each species.

**2.2.2 Animals versus backgrounds.** Participants rarely fixated the background, with normalized fixation proportions to lions (M = .993, SD = .011, CI = .993-.994) and impalas (M = .990, SD = .015, CI = .989-.990) greater than .99. Therefore, although the stimulus species x region interaction was significant, $F_{1.35,43.21} = 50.9$, $p < .001$, the lack of variability renders interpretation inappropriate.

**2.2.3 Specific animal ROIs.** A significant three-way interaction among stimulus species, gaze direction, and ROI suggested that the overall effects of ROI and gaze direction on gaze behavior were affected by the stimulus species presented in the image (Fig 2, Table 1). As such, we reported and interpreted planned contrasts tailored to our specific predictions. Participants fixated the eyes of lions and impalas more than their heads (lion: $p < .001$; impala: $p < .001$) and bodies (lion: $p < .001$; impala: $p < .001$). Participants fixated lion eyes significantly more than the impala eyes ($p < .001$) and directed lions eyes significantly more than averted lion eyes ($p < .001$), but fixated directed and averted impala eyes in similar proportions ($p = .021$).

**2.2.4 Initial fixations on animals.** Participants were more likely to initially fixate the eyes of lions than their heads ($p < .001$) but they initially fixated the eyes of impala in similar proportions as their heads ($p = .44$). Similar to the pattern of results for normalized fixation proportions, participants initially fixated the directed lion eyes at higher proportions than the averted lion eyes ($p < .001$) and they initially fixated the lion eyes more than the impala eyes ($p < .001$). Gaze direction had no effect on initial fixations to impala eyes, (p = .32; Table 2; Fig 3).

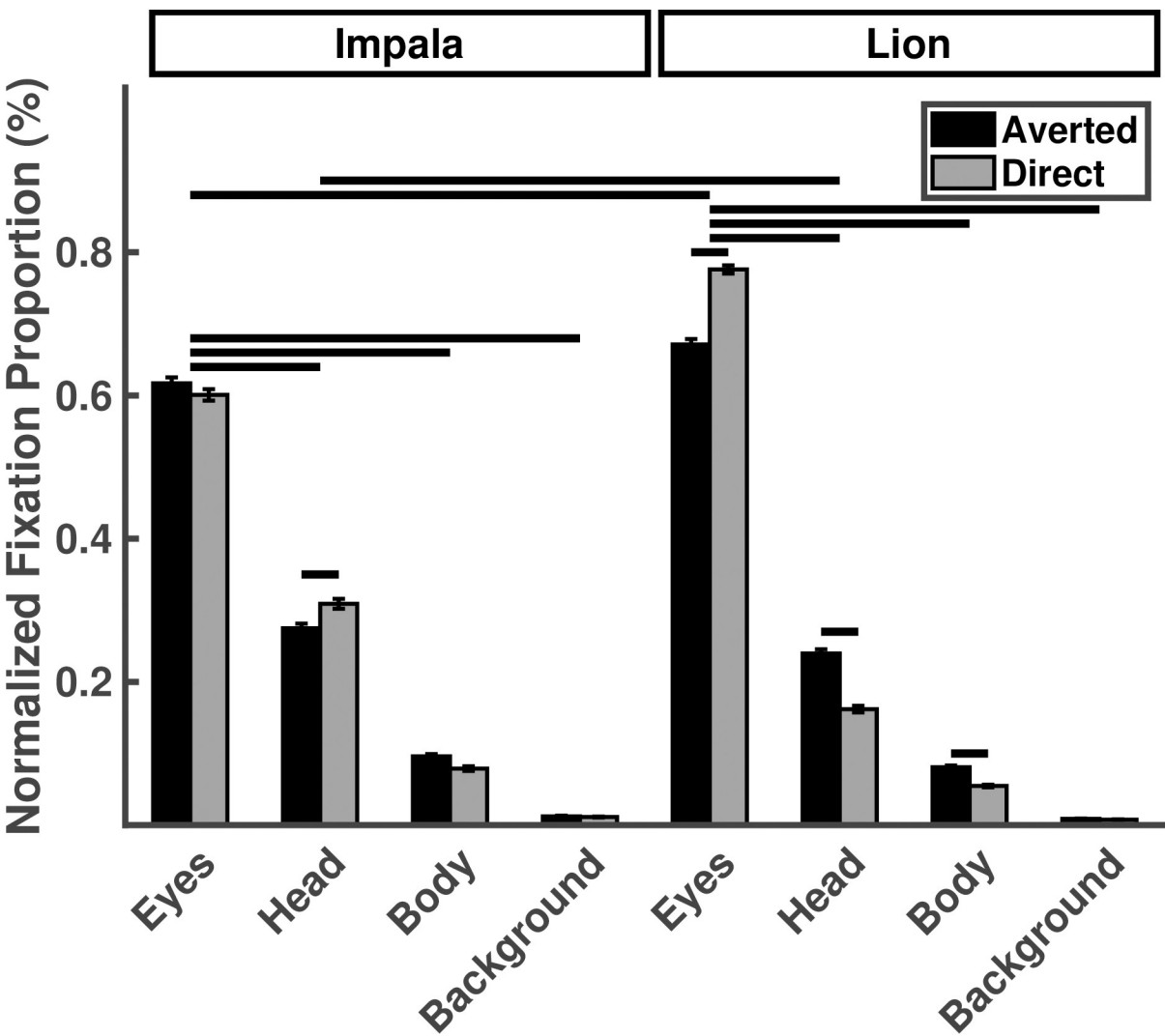

**Fig 2. Normalized fixation proportions plotted as a function of stimulus species, gaze direction and ROI for Experiment 1.** Horizontal lines indicate statistically significant planned contrasts.

**2.2.5 Salience analysis.** The salience at the location of first fixation was compared to the two chance-based estimates described earlier (uniform-random and biased-random). To determine whether the salience model accounted for first fixation position above what would be expected by chance, non-parametric statistics (Mann–Whitney U tests) were performed to compare the medians of fixated salience and uniform-random salience as well as the medians of fixated salience and biased-random salience, with an alpha level of .05. The fixated salience was very low (0.023), as was uniform-random saliency (0.013), U-test fixation salience vs uniform-random salience: $z = 1.645$, $p = 0.100$. Fixated saliency was also no different from biased-random saliency (0.018; U-test fixation saliency vs biased-random saliency: $z = 0.596$, $p = 0.55$). Thus, the salience at fixated locations was no higher than would be expected by the random models.

**2.2.6 Three-way interaction comparisons.** Because the significant three-way interaction among stimulus species, gaze direction, and ROI presented in Table 1 above suggests the presence of significant effects beyond the planned contrasts that were targeted to our specific

**Table 1. The effect of stimulus species, gaze direction, and ROI on the normalized fixation proportions in Experiment 1.** Asterisks indicate statistically significant variables or comparisons.

| Overall Model | | | |
|---|---|---|---|
| | Factor | $F_{df, df\ error}$ | $p$ |
| | Stimulus species | $0_{1,32}$ | >0.99 |
| | Gaze direction | $0_{1,32}$ | >0.99 |
| | ROI | $280.82_{1.08, 34.54}$ | < .001* |
| | Stimulus species* Gaze direction | $0_{1,32}$ | >0.99 |
| | Stimulus species* ROI | $50.90_{1.35, 43.21}$ | < .001* |
| | Gaze direction*ROI | $22.14_{1.21, 38.63}$ | < .001* |
| | Stimulus species* Gaze direction* ROI | $54.19_{1.16, 37.19}$ | < .001* |
| Comparisons | | | |
| | Lion | | |
| | Eyes Directed vs. Eyes Averted | | < .001* |
| | Head Directed vs. Head Averted | | < .001* |
| | Body Directed vs. Body Averted | | .001* |
| | Eyes vs. Head | | < .001* |
| | Eyes vs. Body | | < .001* |
| | Eyes vs. Background | | < .001* |
| | Impala | | |
| | Eyes Directed vs. Eyes Averted | | .021 |
| | Head Directed vs. Head Averted | | < .001* |
| | Body Directed vs. Body Averted | | .036 |
| | Eyes vs. Head | | < .001* |
| | Eyes vs. Body | | < .001* |
| | Eyes vs. Background | | < .001* |
| | Eyes | | |
| | Lion vs. Impala | | < .001* |
| | Head | | |
| | Lion vs. Impala | | < .001* |
| | Body | | |
| | Lion vs. Impala | | .063 |

hypotheses, we conducted unplanned, exploratory, *post hoc* comparisons of the full matrix of three-way contrasts. Bonferroni-adjusted *p* values suggest significant differences in area-normalized fixation proportions between most combinations of the three factors. For example, as an extension of our third prediction regarding the joint influences of stimulus species and gaze direction towards eye ROIs, the three-way comparisons suggest that directed lion eyes are significantly greater than not only averted lion eyes, but also every other combination of the three factors (all *p* < .001). The complete matrix of *post hoc* comparisons is available in S1 Fig and S1 Table. Note that the proportional nature of this measure makes some inferences from lower priority ROIs fraught: an increase in fixations on directed lion eyes, for example, necessarily leads to a harmonic decrease in the fixation proportions to directed lion bodies, heads, and backgrounds.

## 2.3 Experiment 1 discussion

We aimed to test four main predictions related to how humans attend to nonhuman animals, particularly predators and prey. First, drawing from earlier research suggesting an attentional priority to threatening animals [10, 13, 15, 17], we predicted that participants would generally

**Table 2. The effect of stimulus species, gaze direction, and ROI on the initial fixation proportions.** Asterisks indicate statistically significant variables or comparisons.

| Overall Model | | | |
|---|---|---|---|
| | Factor | $F_{df, df\ error}$ | $p$ |
| | Stimulus species | $0_{1,32}$ | >0.99 |
| | Gaze direction | $0_{1,32}$ | >0.99 |
| | ROI | $58.54_{1.43,\ 45.78}$ | < .001* |
| | Stimulus species* Gaze direction | $0_{1,32}$ | >0.99 |
| | Stimulus species* ROI | $31.67_{1.98,\ 63.38}$ | < .001* |
| | Gaze direction*ROI | $15.60_{2.33,\ 74.66}$ | < .001* |
| | Stimulus species* Gaze direction* ROI | $19.25_{1.60,\ 51.09}$ | < .001* |
| Comparisons | | | |
| | Lion | | |
| | Eyes Directed vs. Eyes Averted | | < .001* |
| | Head Directed vs. Head Averted | | < .001* |
| | Body Directed vs. Body Averted | | .82 |
| | Eyes vs. Head | | < .001* |
| | Eyes vs. Body | | < .001* |
| | Eyes vs. Background | | < .001* |
| | Impala | | |
| | Eyes Directed vs. Eyes Averted | | .32 |
| | Head Directed vs. Head Averted | | .002* |
| | Body Directed vs. Body Averted | | .002* |
| | Eyes vs. Head | | .44 |
| | Eyes vs. Body | | < .001* |
| | Eyes vs. Background | | < .001* |
| | Eyes | | |
| | Lion vs. Impala | | < .001* |
| | Head | | |
| | Lion vs. Impala | | .27 |
| | Body | | |
| | Lion vs. Impala | | < .001* |

look at lions more than impala. Because participants gazed nearly exclusively at both lion and impala (and rarely gazed at the background), we could not examine this hypothesis.

Second, we predicted that the eyes of non-human animals would be the most fixated region in natural images. The eyes of both lions and impala were fixated by participants significantly more than the heads, bodies, and background. Initial fixation data also suggested a strong bias for gaze information by human participants: they were more likely to initially fixate the eyes and heads of animals over any other region. These results dovetail with previous literature demonstrating an early and robust preference to fixate the eyes of other people [33, 34, 57], although it should be noted that patterns of gaze behavior to conspecific and heterospecific targets may involve different attentional mechanisms [71]. Viewing patterns were also consistent with previous eye-tracking experiments using animal line illustrations as stimuli [57].

Third, we predicted that fixations of the eyes of nonhuman animals would be influenced by both stimulus species (predator or prey) and the gaze direction of the animal. In support of this prediction, we found that the eyes of lions were looked at significantly more than the eyes of impala. Moreover, we found that the eyes of forward-facing lions were fixated significantly more often (i.e., in normalized fixation proportions) and earlier (i.e., in initial fixations) than the eyes of averted-facing lions in both the humans. Previous studies have reported that

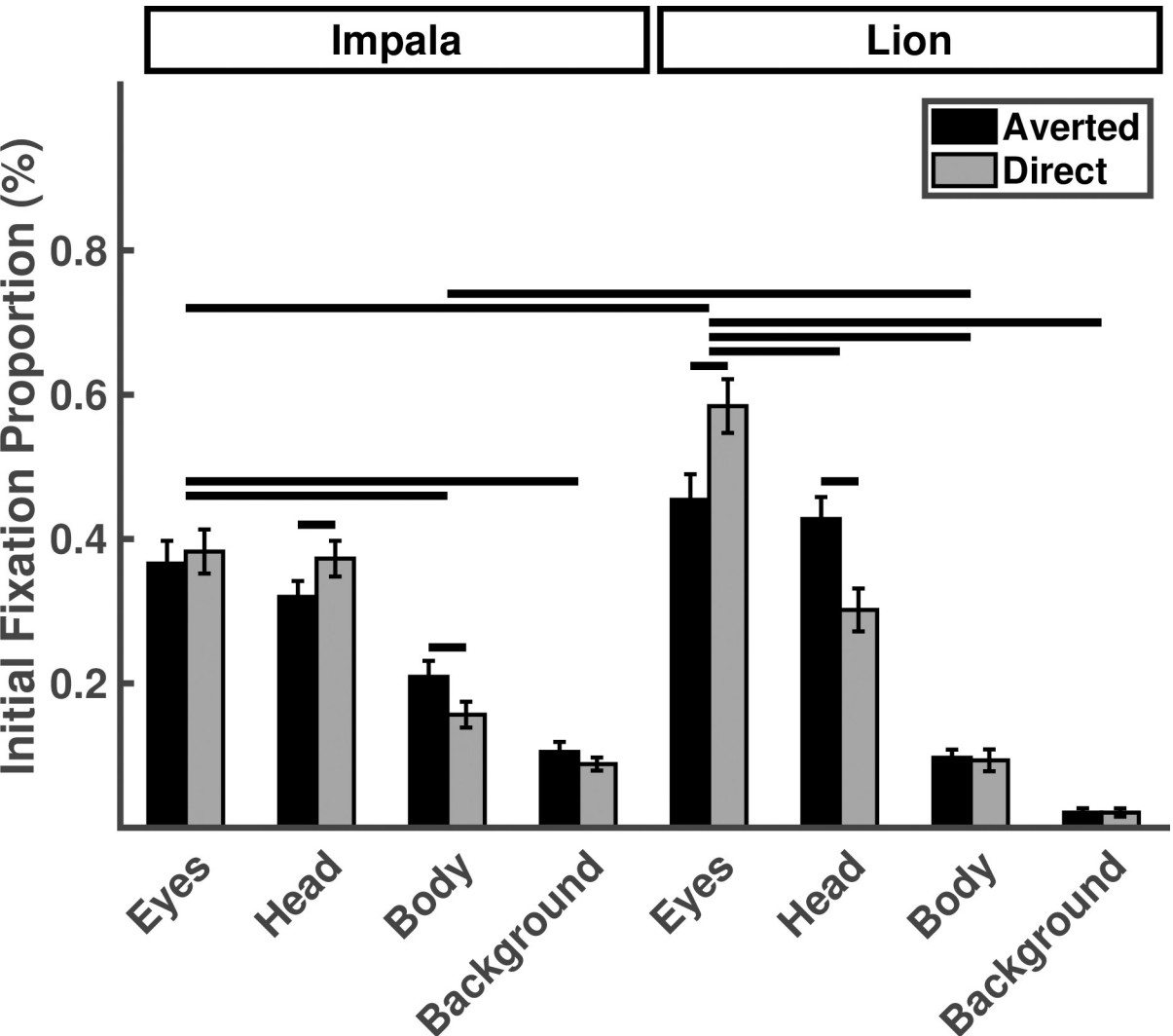

**Fig 3. Proportion of initial fixations falling on eyes, heads, bodies, or background, as a function of stimulus species and gaze direction.** Horizontal lines indicate statistically significant planned contrasts.

forward-facing predators are detected faster relative to averted-facing predators [52]. Our results extend these findings by suggesting that it is the eyes of forward-facing predators, in particular, that may be driving this effect.

Finally, we predicted that bottom-up processing would not account for gaze patterns while viewing animals. Our results showed that the low-level markers of visual saliency we measured using the Salience Toolbox [62] offered a weak explanation for the data. Instead, it appears that attention is likely directed toward the eyes of animals as a result of their importance in assessing risk.

## 3. Experiment 2

### 3.1 Experiment 2 methods

**3.1.1 Participants.** Seven group-housed adult common chimpanzees (4 females and 3 males; aged 29–51 years, $M$ = 41 years) at The University of Texas MD Anderson Cancer

Center's Michale E. Keeling Center for Comparative Medicine and Research participated in this experiment between August and October 2022. Two of the chimpanzees (Sandy, Simba) were born in the wild in regions distant from savannas or lion habitats. The other five chimpanzees (Bo, Chechekul, Lulu, Mandy, and Punch) were born in captivity.

Chimpanzees participated in the research voluntarily, choosing to interact with the experimenter and apparatus at their sole discretion. Both during experimental sessions and at all other times the chimpanzees had full access to indoor areas, outdoor areas, conspecifics (2–4 for each chimpanzee in this experiment), food (fresh fruits, vegetables, and commercial primate chow), clean water, and enrichment. These chimpanzees were highly familiar with humans and cognitive-behavioral tasks, although previous exposure to computers and eye-trackers was minimal. The chimpanzees exhibited no signs of distress during the experiment. All research was approved by the Institutional Animal Care and Use Committee of the Keeling Center (IACUC # 0894-RN01) and Texas A&M University (IACUC # 2022–0089 EX), followed the guidelines of the Institute of Medicine on the use of chimpanzees in research, complied with the Society for Neuroscience Policy on Ethics, and reported in accordance with ARRIVE guidelines.

**3.1.2 Apparatus.** The Tobii TX300 eye-tracker (with built-in monitor) was used for all chimpanzee testing. To properly position the eye-tracker in front of the steel mesh of the animals' enclosures, we affixed the eye-tracker to a rolling utility cart at roughly the height of a sitting, adult chimpanzee (approximately 70 cm). To incentivize the chimpanzees to properly position themselves at the eye-tracker, we affixed a polycarbonate panel connected to a peristaltic dosing pump to the mesh to reward chimpanzees with fruit-flavored, sugar-free drink mix (hereafter: juice) for their participation in the experiment (S2 Fig). Python software was used to control stimulus presentation (via the pygame library), eye-tracker data collection (via Tobii's Python Software Development Kit), and the juice pump (via a USB relay controlled with Python's serial module). We applied the noise-robust fixation classification algorithm I2MC [72] to classify chimpanzee gaze data as fixations or saccades; this fixation filter is robust to noise and data loss.

**3.1.3 Procedure.** Chimpanzees completed this experiment in the indoor portion of their normal housing between 12:00 and 15:00. To calibrate the eye-tracker before each experimental session, we prepared a custom shaping and 2-point calibration procedure. First, chimpanzees were rewarded with juice for looking at any location on the monitor while the monitor was displaying a large (1080 x 1080 pixels) video. These were publicly available videos of monkeys feeding or playing, selected to attract chimpanzee gaze to the screen (downloaded from pixabay.com). Next, chimpanzees were rewarded only for sustaining gaze to the video for at least 400 msec, and the video incrementally decreased in size until it was sufficiently small to act as a calibration point (300 x 300 pixels). Finally, the primate video moved to the upper-right and then the lower-left quadrants of the screen (or vice versa) to calibrate the chimpanzee's gaze at these two points according to the calibration thresholds set by Tobii's 2-point calibration procedure. For experienced chimpanzees, this calibration process took a few minutes. Based on our validation procedure (S1 File), the accuracy of the eye-tracker using chimpanzee subjects was approximately 1.5˚ (this level of accuracy is similar to other eye-tracking studies using chimpanzees [42]).

After calibration, chimpanzees were presented with their first trial. A central fixation cross appeared. If the chimpanzee fixated within 100 pixels of this point for a duration of >50 ms, the software automatically presented one of the lion or impala stimuli for 3 sec. This duration was shorter than the 8 sec stimulus presentations of Experiment 1 and was instead the duration used in many previous great ape eye-tracking experiments (e.g., [44, 73–77]). While the image was onscreen, the dosing pump dispensed juice rewards continuously to the chimpanzee as

long as their gaze remained onscreen. Although several reward schedules have been used in ape eye-tracking designs (e.g., every 3 seconds of activity [78]; at the start of trials, [74]), continuous juice rewards were both more appropriate for testing animals in their home enclosures (alongside conspecifics, husbandry staff, other experimenters, and a highly-enriched captive setting) and unlikely to engender specific biases of gaze behavior (e.g., to directed lion heads but averted impala backgrounds). The apes were not explicitly trained to view the stimuli in any way.

At the conclusion of the trial, the fixation screen returned to prompt the chimpanzee to begin the next trial. To minimize any biasing effects of the chimpanzees' environment (e.g., groupmates, activity in nearby chimpanzee groups, human experimenter or caretaker vocalizations or behaviors), each of the 96 images was shown to each chimpanzee four times in randomized order (thus, each chimpanzee viewed 384 images). If the chimpanzee did not gaze onscreen for at least 10% of a trial duration, the trial was omitted from the analysis and the trial was repeated later in the session. Stimulus presentations were randomized. Because the chimpanzees were in control of each eye-tracking session's duration, some chimpanzees finished all trials in a single day whereas other chimpanzees took as many as 4 days to complete testing.

**3.1.4 Stimuli.** The same set of 96 digital images of male lions and female impala were used in Experiment 2. Because of differences in screen resolution between the human (1024 x 768 pixels) and chimpanzee (1920 x 1080 pixels) eye-tracker monitors, the photographs were presented slightly differently between experiments. In particular, whereas the photographs filled the entire screen (1024 x 768 pixels) in the experiments with human subjects the photographs were centered and nearly filled the entire screen (1600 x 1080 pixels) for the experiments with the chimpanzee subjects.

**3.1.5 Eye-tracking analysis.** Eye-tracking analyses were the same as in Experiment 1.

**3.1.6 Salience analysis.** Salience analysis were the same as in Experiment 1, this time using chimpanzee fixation data to calculate median salience at fixation and the two control variables (uniform-random salience and biased-random salience).

## 3.2 Experiment 2 results

**3.2.1 Statistical analysis.** Statistical analyses were the same as in Experiment 1. All analyses were performed for the human and chimpanzee data because the experimental designs were not identical. Because fixation duration is more typically used as the dependent measure in eye-tracking designs with nonhuman primates, alternative analyses with fixation durations are presented in Supplemental Analyses S4 Fig.

**3.2.2 Animals versus backgrounds.** Chimpanzee gaze behaviors were biased to lions (M = .91, SD = .10; 95% CI = .84-.98) and impala (M = .84, SD = .14; 95% CI = .74-.94) versus the image backgrounds, stimulus species x region $F_{1,6}$ = 27.57, p = .002. Chimpanzees fixation proportions to the lions relative to their backgrounds were significantly higher than fixation proportions to impala relative to their backgrounds (p < .001).

**3.2.3 Specific animal ROIs.** Due to a significant three-way interaction among stimulus species, gaze direction, and ROI we reported and interpreted planned contrasts tailored to our specific predictions (Fig 4, Table 3). Chimpanzees fixated the eyes at similar proportions to heads and bodies of lions (eyes vs. heads: $p$ = .95; eyes vs. bodies: $p$ = .17) and fixated the eyes of impalas at similar proportions to their heads ($p$ = .71), but fixated impala eyes less than their bodies ($p$ < .001). fixated the eyes of lions and impalas more than their heads (lion: $p$ < .001; impala: $p$ < .001) and bodies (lion: $p$ < .001; impala: $p$ < .001). Chimpanzees fixated lion eyes significantly more than the impala eyes ($p$ < .001) and directed lions eyes significantly more

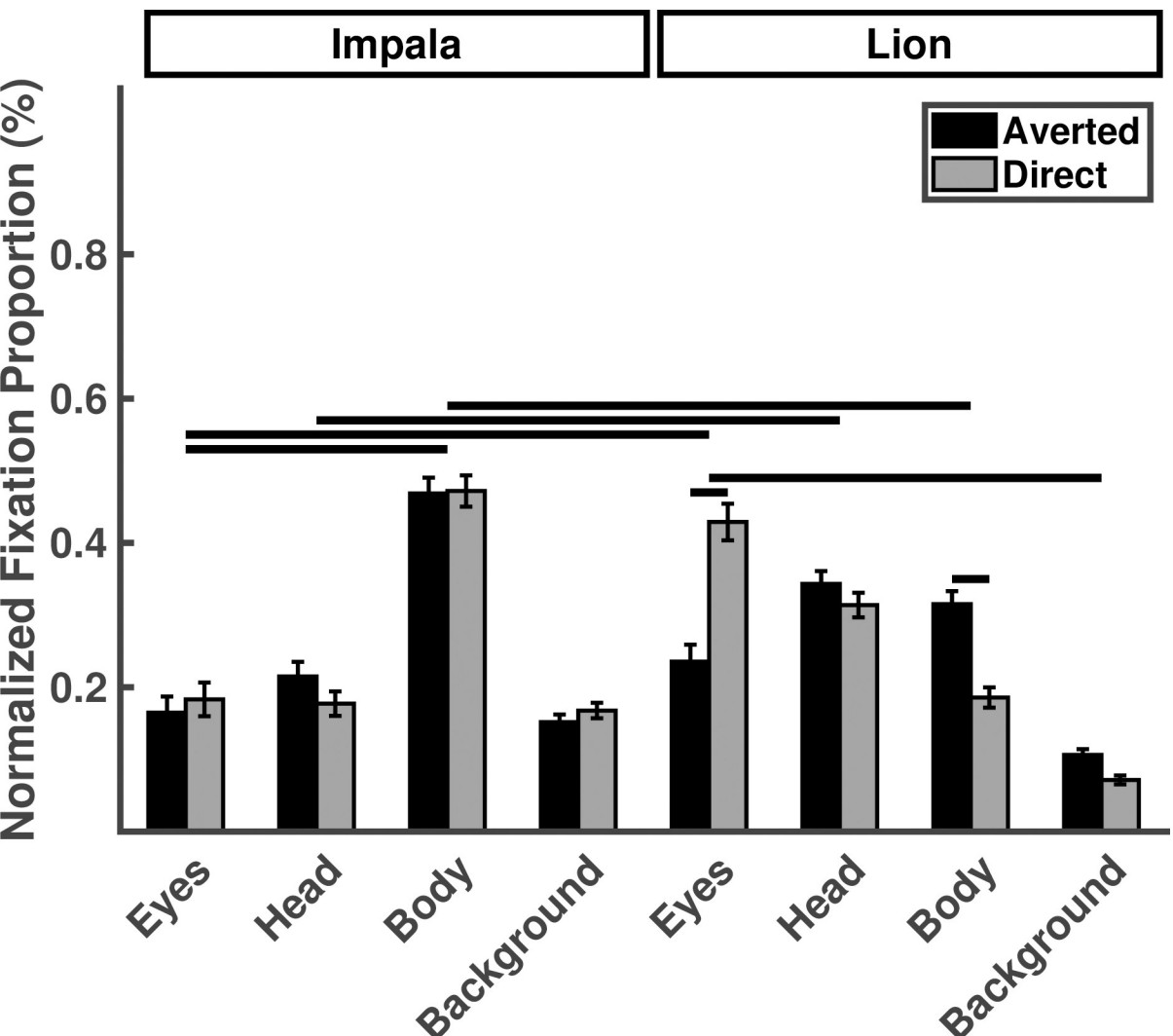

**Fig 4. Normalized fixation proportions plotted as a function of stimulus species, gaze direction and ROI for Experiment 2.** Horizontal lines indicate statistically significant planned contrasts.

than averted lion eyes ($p < .001$), but fixated directed and averted impala eyes in similar proportions ($p = .47$). These effects were generally consistent across the gaze behaviors of individual chimpanzees (S3 Fig), and were maintained when area-normalized fixation duration was taken as the dependent variable instead of area-normalized fixation counts used in these results (S4 Fig).

**3.2.4 Initial fixations on animals.** Chimpanzees initially fixated the eyes of lions at lower proportions compared to their heads ($p < .001$), but also initially fixated the eyes of impala in similar proportions to their heads ($p = .67$). Chimpanzees initially fixated the directed lion eyes at higher proportions than the averted lion eyes ($p = .002$) and they initially fixated the lion eyes more than the impala eyes (chimpanzees: $p = .03$). Gaze direction had no effect on initial fixations to impala eyes $p = .89$; see Table 4; Fig 5).

**3.2.5 Salience analysis.** As in Experiment 1, fixated salience was very low (0.017), as was uniform-random saliency (0.013); U-test fixation salience vs. uniform-random salience: $z = 1.788$, $p = 0.074$. Fixated salience was also no different from biased-random saliency (0.019;

**Table 3. The effect of stimulus species, gaze direction, and ROI on the normalized fixation proportions in Experiment 2.** Asterisks indicate statistically significant variables or comparisons.

| Overall Model | | | |
|---|---|---|---|
| | **Factor** | $F_{df, df\ error}$ | $p$ |
| | Stimulus species | $0_{1,6}$ | >0.99 |
| | Gaze direction | $0_{1,6}$ | >0.99 |
| | ROI | $6.88_{1.71,\ 10.25}$ | .02 |
| | Stimulus species* Gaze direction | $0_{1,6}$ | >0.99 |
| | Stimulus species* ROI | $25.63_{3,\ 18}$ | < .001* |
| | Gaze direction*ROI | $16.44_{3,\ 18}$ | < .001* |
| | Stimulus species* Gaze direction* ROI | $8.59_{3,\ 18}$ | < .001* |
| **Comparisons** | | | |
| | Lion | | |
| | Eyes Directed vs. Eyes Averted | | < .001* |
| | Head Directed vs. Head Averted | | .25 |
| | Body Directed vs. Body Averted | | < .001* |
| | Eyes vs. Head | | .95 |
| | Eyes vs. Body | | .17 |
| | Eyes vs. Background | | < .001* |
| | Impala | | |
| | Eyes Directed vs. Eyes Averted | | .47 |
| | Head Directed vs. Head Averted | | .14 |
| | Body Directed vs. Body Averted | | .90 |
| | Eyes vs. Head | | .71 |
| | Eyes vs. Body | | < .001* |
| | Eyes vs. Background | | .81 |
| | Eyes | | |
| | Lion vs. Impala | | < .001* |
| | Head | | |
| | Lion vs. Impala | | < .001* |
| | Body | | |
| | Lion vs. Impala | | < .001* |

U-test fixation saliency vs biased-random saliency: 0.869, z = -0.165). Thus, the salience at locations fixated by the chimpanzees was no higher than would be expected by the random models.

**3.2.6 Three-way interaction comparisons.** As in Experiment 1, the significant three-way interaction among stimulus species, gaze direction, and ROI suggests significant additional effects beyond those of the planned contrasts of our explicit hypotheses. Bonferroni-corrected *p* values over the complete matrix of post hoc comparisons again suggest that directed lion eyes were the target of greater fixation proportions than averted lion eyes (*p* < .001) as well as directed lion heads (*p* < .05), bodies (*p* < .001), or backgrounds (*p* < .001). The complete matrix of *post hoc* comparisons is available in S5 Fig and S2 Table.

### 3.3 Experiment 2 general discussion

Experiment 2 tested in chimpanzees the same four predictions for anticipated patterns of gaze behavior to lion and impala eyes and faces with directed or averted orientations as were tested in humans in Experiment 1. First, we predicted that chimpanzees would look more at lions than impala and found that they reliably looked more at lions than at impala (versus their

**Table 4. The effect of stimulus species, gaze direction, and ROI on the initial fixation proportions in Experiment 2.** Asterisks indicate statistically significant variables or comparisons.

| Overall Model | | | |
|---|---|---|---|
| | Factor | $F_{df, df\ error}$ | *p* |
| | Stimulus species | $0_{1, 6}$ | >0.99 |
| | Gaze direction | $0_{1, 6}$ | >0.99 |
| | ROI | $97.93_{3, 18}$ | < .001* |
| | Stimulus species* Gaze direction | $0_{1, 6}$ | >0.99 |
| | Stimulus species* ROI | $55.93_{3, 18}$ | < .001* |
| | Gaze direction*ROI | $8.95_{3, 18}$ | < .001* |
| | Stimulus species* Gaze direction* ROI | $15.63_{3, 18}$ | < .001* |
| Comparisons | | | |
| | Lion | | |
| | Eyes Directed vs. Eyes Averted | | .002* |
| | Head Directed vs. Head Averted | | < .001* |
| | Body Directed vs. Body Averted | | .07 |
| | Eyes vs. Head | | < .001* |
| | Eyes vs. Body | | < .001* |
| | Eyes vs. Background | | < .001* |
| | Impala | | |
| | Eyes Directed vs. Eyes Averted | | .89 |
| | Head Directed vs. Head Averted | | .64 |
| | Body Directed vs. Body Averted | | .07 |
| | Eyes vs. Head | | .67 |
| | Eyes vs. Body | | < .001* |
| | Eyes vs. Background | | < .001* |
| | Eyes | | |
| | Lion vs. Impala | | .03 |
| | Head | | |
| | Lion vs. Impala | | < .001* |
| | Body | | |
| | Lion vs. Impala | | .18 |

backgrounds). This result supports our first prediction and contributes to previous work suggesting an attentional bias for dangerous animals [10, 20, 79].

Second, we predicted that chimpanzees would fixate the heterospecific eyes more than any other region. Chimpanzees did not exhibit these gaze patterns. The chimpanzees did not fixate the eyes of the lions and impalas more than their heads and bodies, nor were their initial fixations biased toward the eyes. Although this prediction was not supported, it coheres with previous research with nonhuman primates that suggests less intense bias(es) to conspecific and heterospecific eyes in great apes, especially chimpanzees [78].

Third, we predicted that fixations to the lion and impala images would be influenced by the animal type and the gaze direction of the animal. This prediction was broadly supported: lion eyes were more fixated than impala eyes, and forward-facing lion eyes were fixated at higher proportions of overall fixations and initial fixations than the averted-facing lion eyes. Although chimpanzees were not as attentive to eyes overall (prediction 2), they nevertheless attended to eyes, predators, and orientation in ways that suggest broad biases to forward-facing predators like those that have been previously reported in other eye-tracking designs with humans [52] and a special role for visible eyes in these biases. That this pattern of attention to animal eyes

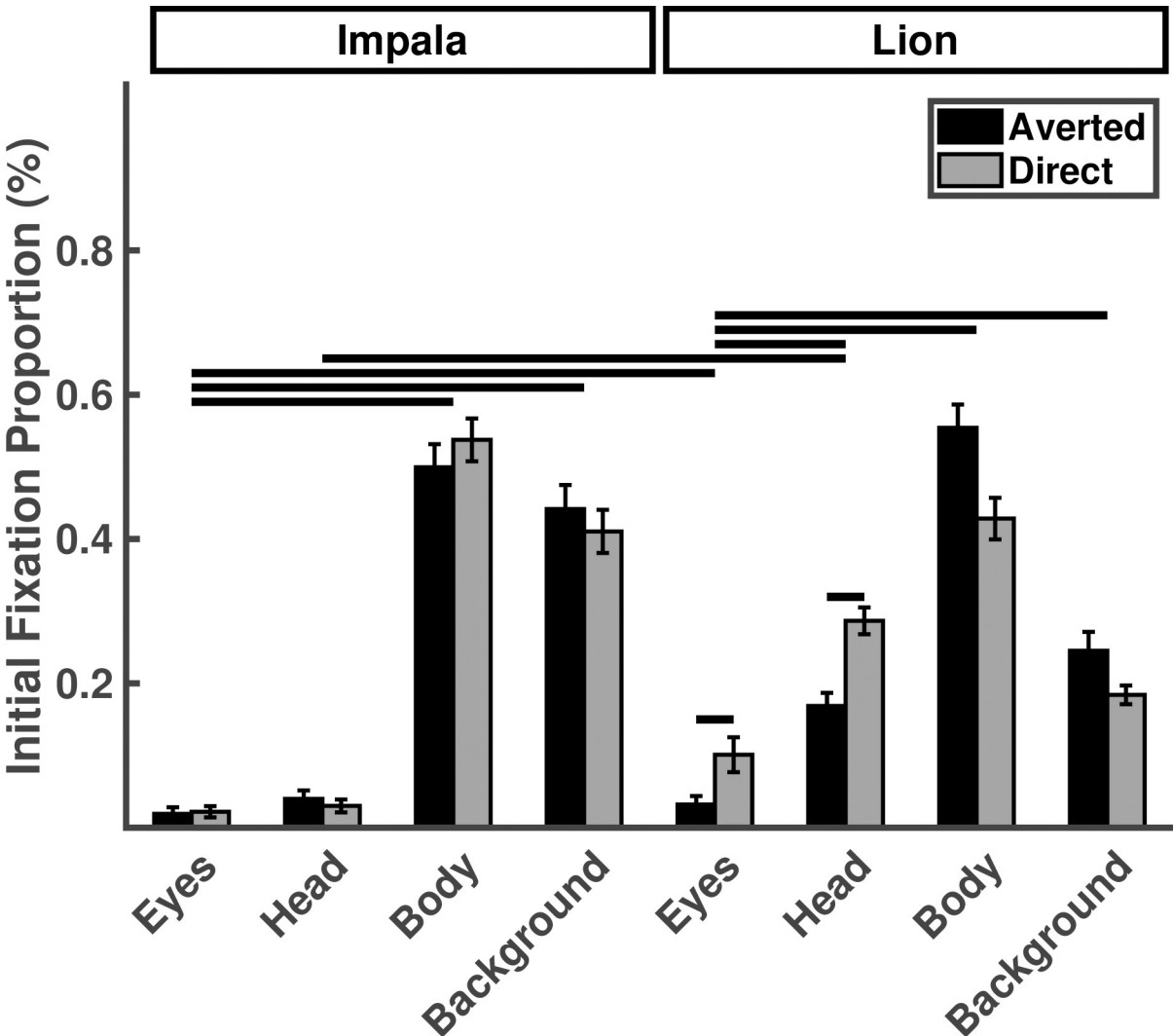

**Fig 5. Proportion of initial fixations falling on eyes, heads, bodies, or background, as a function of stimulus species and gaze direction for Experiment 2.** Horizontal lines indicate statistically significant planned contrasts.

extended to naïve chimpanzees without prior exposure to lions, impala, or images of predators (with the possible exception of the two wild-caught chimpanzees) further emphasizes the more evolutionarily-conserved, goal-directed nature of attention to predators and prey.

Finally, salience modeling did not suggest that chimpanzee gaze behaviors can be understood as a mere recapitulation of biases to low-level stimulus features.

## 4. Discussion

Eyes are a rich source of information that can convey the level of threat posed by potentially dangerous predators [80, 81]. Across two designs with humans and chimpanzees, we tested several predictions about attentional biases to eyes of prototypical predators (male lions) and prototypical prey (female impala). Our predictions about the gaze patterns of the chimpanzees generally matched those of the humans in ways that replicate and extend what is currently known about the attentional priorities of primates. Chimpanzees attended more to lions than impala (supporting prediction 1) and attended to the eyes of forward-facing lions more than

the eyes of averted lions (supporting prediction 3) in a way seemingly divorced from low-level stimulus information (supporting prediction 4). To our knowledge, these experiments provide some of the first evidence examining the strength of the bias towards visually fixating the eyes of heterospecific nonhuman animals in both human and chimpanzee participants.

Several design considerations complicate such direct comparison between the patterns of gaze behavior of humans and chimpanzees. Chimpanzees were exposed to multiple repetitions of the images, in shorter durations, among other typical compromises of working with the laboratory-housed population (e.g., nearby conspecifics, juice rewards, specific hardware/software). In addition, human participants were familiarized to lions by semantic learning, popular media, and lived experience (e.g., a trip to the zoo) in a way that the chimpanzees were not. For example, a major discrepancy between chimpanzee and human gaze patterns was of overall attention to eyes (prediction 2), which were the most fixated region for humans but not for chimpanzees. One possibility is that the chimpanzees disengaged attention from the eyes to explore the rest of the images more fully compared with the human participants (as in [42, 74]) or, as referenced above, it may be that humans' prior learning and enculturation (e.g., already knowing what an impala looks like) biased their gaze behavior away from what the chimpanzees were likely attending to for the first time. Whether this sort of discrepancy should be understood as species difference, design difference, difference in enculturation, or some confluence of all of the above cannot be adequately answered using only the pair of experiments we presented and represents a target for future research.

In addition to narrowing the eye-tracking design differences between the two species, future research might measure similar attentional biases in other primate species, like bonobos, which have a similar evolutionary history to chimpanzees but have previously demonstrated divergent patterns of attention to conspecific faces [78]. Future designs may also test the prediction of attentional biases to images of dangerous animals by presenting them alongside images of less dangerous animals, using a preferential viewing design, and present a larger diversity of predator and non-predator species to better describe the features of heterospecific animals that elicit these attentional biases in primates.

In conclusion, both humans and chimpanzees exhibited a strong preference for visually fixating the eyes of forward-facing predators. Because the salience modeling predicted that low-level, visually salient features would produce very different gaze patterns than those produced by either the human or chimpanzee participants, low-level features are a weak explanation for the observed viewing patterns. Although previous learning about lions and their behaviors could have led human participants to look to the lions' eyes and faces, the chimpanzees were naïve to images of lions, had no such learning, and nevertheless looked to lion eyes in ways that distinctly yet equivalently accorded with our predictions. Instead, a more likely explanation for our results could be that humans and chimpanzees fixate the two facing eyes of animals in ways that reflect their visual properties due to the adaptive significance of assessing the focal interests of these animals and their likelihood of posing significant threats.

## Supporting information

**S1 File. The procedure for validating chimpanzee eye-tracker calibrations.**
(DOCX)

**S1 Table. *p* values for *post hoc* comparison matrix for Experiment 1.**
(DOCX)

**S2 Table. *p* values for *post hoc* comparison matrix for Experiment 2.**
(DOCX)

**S1 Fig.** *Post hoc* **comparison matrix for Experiment 1.**
(DOCX)

**S2 Fig. The apparatus used for testing chimpanzee subjects.** (A) the eye-tracker and all associated hardware positioned on a rolling cart approximately 63 cm from the chimpanzee mesh, and (B) an overhead view of a chimpanzee participating in the experiment. Photograph taken by WW and printed under a CC BY license.
(DOCX)

**S3 Fig. Area-normalized fixation proportions for individual chimpanzee subjects.**
(DOCX)

**S4 Fig. Supplemental analyses.** Fixation duration plotted as a function of stimulus species, gaze direction and ROI for chimpanzee subjects.
(DOCX)

**S5 Fig.** *Post hoc* **comparison matrix for Experiment 2.**
(DOCX)

## Author Contributions

**Conceptualization:** Alan Kingstone, Richard Coss, Elina Birmingham, Jessica L. Yorzinski.

**Data curation:** Will Whitham.

**Formal analysis:** Will Whitham, Nicola C. Anderson, Walter F. Bischof.

**Funding acquisition:** Alan Kingstone, Elina Birmingham, Jessica L. Yorzinski.

**Investigation:** Will Whitham, Bradley Karstadt, Nicola C. Anderson, Alan Kingstone, Elina Birmingham.

**Methodology:** Will Whitham, Nicola C. Anderson, Walter F. Bischof, Elina Birmingham, Jessica L. Yorzinski.

**Project administration:** Steven J. Schapiro, Elina Birmingham, Jessica L. Yorzinski.

**Software:** Will Whitham.

**Supervision:** Nicola C. Anderson, Steven J. Schapiro, Alan Kingstone, Elina Birmingham, Jessica L. Yorzinski.

**Writing – original draft:** Will Whitham, Bradley Karstadt, Nicola C. Anderson, Alan Kingstone, Elina Birmingham, Jessica L. Yorzinski.

**Writing – review & editing:** Will Whitham, Bradley Karstadt, Nicola C. Anderson, Walter F. Bischof, Steven J. Schapiro, Alan Kingstone, Richard Coss, Elina Birmingham, Jessica L. Yorzinski.

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
