## [Decision Letter · Decision Letter 0]

20 Feb 2024

PONE-D-23-40092Predator gaze captures both human and chimpanzee attentionPLOS ONE

Dear Dr. Whitham,

Thank you for submitting your manuscript to PLOS ONE. After careful consideration, we feel that it has merit but does not fully meet PLOS ONE’s publication criteria as it currently stands. Therefore, we invite you to submit a revised version of the manuscript that addresses the points raised during the review process.

We look forward to receiving your revised manuscript.

Kind regards,

Nick Fogt

Academic Editor

PLOS ONE

 [This work was supported by the National Science Foundation (BCS #1926327) to W.W., S.S., and J.Y., Natural Sciences and Engineering Research Council of Canada (RGPIN-2022-03079) to B.K., N.A., and A.K].  

6. We note that Figure 1 and S1 in your submission contain copyrighted images. All PLOS content is published under the Creative Commons Attribution License (CC BY 4.0), which means that the manuscript, images, and Supporting Information files will be freely available online, and any third party is permitted to access, download, copy, distribute, and use these materials in any way, even commercially, with proper attribution. For more information, see our copyright guidelines: http://journals.plos.org/plosone/s/licenses-and-copyright.

a. You may seek permission from the original copyright holder of Figure 1 and S1 to publish the content specifically under the CC BY 4.0 license. 

Additional Editor Comments:

Both reviewers appreciate the topic of the paper and acknowledge that studies in this area are needed. Both reviewers have a number of comments. One particular concern (Reviewer #1) is the generalizability of the results. Please address each of the reviewer comments.

Reviewers' comments:

Reviewer's Responses to Questions

**Comments to the Author**

1. Is the manuscript technically sound, and do the data support the conclusions?

Reviewer #1: Partly

Reviewer #2: Yes

2. Has the statistical analysis been performed appropriately and rigorously? 

Reviewer #1: No

Reviewer #2: Yes

3. Have the authors made all data underlying the findings in their manuscript fully available?

Reviewer #1: Yes

Reviewer #2: No

4. Is the manuscript presented in an intelligible fashion and written in standard English?

Reviewer #1: Yes

Reviewer #2: Yes

5. Review Comments to the Author

Reviewer #1: This is an interesting and well-written manuscript about a topic that has not attracted much prior research and as such is welcomed. There were however some important deficiencies in theoretical justification and experimental design (or insufficient explanation) that need to be addressed. Perhaps the authors have already thought through and solved many of these, but the reasoning should be included in the manuscript for readers to follow all points of their logic.

Major concerns:

The paper refers throughout to predator and prey gaze, but tests only lion and impala with no other species. This seriously limits the generalizability of their conclusions. This is compounded by fairly weak justification of species choice. It is true lions co-occur with eastern chimpanzees in some narrow savannah regions where chimpanzees are likely hunted to at least some extent, but most captive chimpanzees, and presumably the current population, are western chimpanzees. Beyond subspecies differences, most chimpanzees live in forest environments. For both of these reasons, leopards are much more ecologically relevant and indeed have been found to predate chimpanzees across their range more than has been reported for lions. Further, the stimuli examples appeared to show male lions, who do not typically engage in much of the hunting, and lions further are group-hunters (not solitary as are leopards) and only one individual at a time was presented. While the big cat morphology may be a form that is relatively conserved, and predates upon many primates across the world including humans and chimpanzees, the text only makes a very vague association with one species in one part of a different subspecies’ range. I recommend justifying this point in the text, that we may expect conserved attentional biases to come through across big cat morphologies including the lions used in this study. Relatedly, on the other side, chimpanzees are proficient hunters themselves but do not hunt impala. Duikers, squirrels, red colobus monkeys, would all be more species-relevant prey to a chimpanzee, why was impala chosen? This should also be mentioned when discussing why human attention may differ from other species. Chimpanzees, like humans, are both predators are prey to different animals, but the stimuli only included species that are predator and prey to humans. It is thus not really a fair species comparison. The discussion certainly should mention the lack of species diversity as a limitation on generalizability and future direction, even moreso in light of these larger concerns.

The inclusion of the saliency maps felt a little out of place and I did not find them integrated into the rest of the discussion. Some introductions of their actual use and predictive power in the introduction would be of use. As of now, they are introduced only in ways they fail to predict gaze, and then the results again fail to find them making new predictions, and the naïve reader is left wondering what these can contribute if they consistently fail to make predictions.

The hypotheses are quite vague and not well-justified, which combined with a very high number of statistical tests leads readers to wonder if this was more exploratory or hypothesis-driven. Both are welcomed, especially with new topics such as this, but perhaps the manuscript could benefit from distinguishing them and making clearer, concrete predictions associated with the actual hypotheses (ideally with citation). In particular, the first set of predictions (paragraph starting line 131) is essentially just sameness or difference, but then no statistical tests actually directly compared the two species. The lack of direct human-chimpanzee comparison is a strange choice. The reasoning of the sentence starting on line 135 is especially vague and nebulous. If these were the topics of interest wouldn’t a study comparing humans with different background knowledge and social contexts be a more salient test? The choice of a cross-species framework here could be much better-justified.

The dependent measure as proportion of fixation is very unusual compared to previous great ape work. Why was total fixation duration not included? What reasoning is there for choosing number of fixations over duration, especially considering prior work consistently showing chimpanzee gaze is likely to jump around more than humans. Could the lion have evoked more glances away (chimpanzees tend to avoid prolonged direct eye contact) while the averted gaze and impala had longer but more consistent gaze? At the very least, supplemental analysis detailing the duration should be included somewhere.

Minor concerns:

Line 87: The topic of chimpanzees seems to come out of nowhere here. As they are central to the whole paper, their introduction should be naturally connected. Perhaps a linking phrase of paragraph about the value and perspective around comparative gaze studies of chimpanzees would help improve this and address one of the major comments above?

Line 87: This finding itself also needs more rigour if chimpanzee attention to faces is a central focus of the paper. The previous studies find that chimpanzees gaze more at mouth than eyes even within faces, and this has been considered as potentially more species-relevant for social cues than eyes (Kano et al. 2018, Brooks et al 2021). Did the authors consider looking at attention to mouth by species given this finding? At the very least, the difference in the role and meaning of eye-contact across species should be mentioned. Note also that the supplemental material of some of these previous face-based studies have looked at attention to direct compared to averted gaze, which while not directly on the topic of predator/prey gaze may be useful for the authors to look at.

Line 116: This is a bit of a strange citation here, as the point of the sentence is that eyes have high saliency in the face but the cited paper is on the topic of that not being at all universal, and certainly is not a citation about them being “presented in this manner.”

Line 139: This whole paragraph does not distinguish species for the predictions. Maybe the authors intend these predictions to apply to all participants, but this should then be made explicit.

Line 161: The ethnicity states Chinese, Korean, and east Asian as distinct, please check this or clarify.

Throughout: “other part of the head” may be clearer than just head when distinguishing from eyes, since in some sentences this could easily be lost by readers.

Several times: It may be a bit inaccurate to say we couldn’t look at lion-impala differences in humans because of high attention to both. It would be more accurate to say no differences were observed because of ceiling effects, or the data was not suited for a formal model because of ceiling effects but visually appeared identical. Saying both were at ceiling is a comparison itself.

Line 430: “To our knowledge, this experiment provides some of the first evidence examining the strength of the bias towards visually fixating the eyes of nonhuman animals in both human and chimpanzee participants.” This sentence must be rephrased or removed. The prior sentence cites Kano et al. 2018, which compared several primate species including humans on social attention. Also, the nonhuman animals point is somewhat missed here if talking about nonhuman participants, maybe allospecific would be more relevant.

General: the size of the eye ROIs appears very small in the figure, especially considering the screen size used. Are these within the margin of error of accuracy? This should be mentioned.

Line 443: Chimpanzees are one of the nearest relatives, bonobos are equally closely related. Bonobos generally look much more to eyes than chimpanzees (Kano et al. 2018) and have no lions in their range (but do also face leopard predation) and are less proficient hunters. Perhaps a future comparative approach can be highlighted as a future direction?

Line 444: check divergence dates.

General: the discussion could use more interpretation and connection to the broader literature, as it stands it is hard to see what the importance beyond restatement of the results.

One point not raised anywhere in the manuscript that certainly bears discussion: predators have more forward-facing eyes than prey. Could you rule out the fact that there is a general direct eye-gaze bias, but that the lions just have more direct gaze (at least from the view of primates such as ourselves with forward-facing eyes) and it is therefore not related to predator/prey distinctions per se?

How do you account for the fact that the relative eye size between the two species is different? The eyes of the impala appear to be a much bigger proportion of the head than lions. Perhaps this strengthens the finding, but it bears discussion.

It is rare in eye-tracking with great apes to use a mesh rather than transparent barrier between eyes and screens. Can you be sure this did not affect gaze tracking when one or the other eye is blocked to the sensor?

Similarly, juice only being delivered while gaze is onscreen seems very odd. How can you be sure their behaviour was not simply shaped by perceived rewarding of certain gaze? Generally, juice is used to motivate them to join experiments, but rewards that are differentially delivered according to gaze is a slippery slope if aiming to target spontaneous gaze patterns.

How did attention change across/within trials? The trials seem longer than typical in chimpanzee research, was there any obvious effect of presentation order?

Was there consistency across individuals tested within species? This would strengthen the finding considerably given the low sample size.

There are several citation errors. Ensure the reference section matches the main text (e.g. Carter et al. wrong year, Kano et al. 2018 not in reference section).

Reviewer #2: GENERAL COMMENTS

In this study, Karstadt & Whitham et al. investigate how attentional patterns in humans and chimpanzees are influenced by predation. Utilising eye-tracking technology, they examined how features of potential predator and prey species affect attention, and how the gaze of the target animal modulates this phenomenon. This study is significant as our understanding of attentional mechanisms in response to predators and prey across different species remains limited. I would like to express my gratitude to the authors for their diligent efforts and for presenting a well-written and clear manuscript. I believe it is suitable for publication pending the below suggestions for improving the paper. Firstly, I believe that incorporating additional primate studies would complement the human research and comparative aspect of the manuscript. Secondly, I have some inquiries regarding the data processing and statistical analysis, which I will outline below. I hope these suggestions are constructive and beneficial for the authors.

ABSTRACT

L29: It is not fully clear here what you mean with 'visual features'. Does this include the gaze orientation, or the low-level features which you discuss later on.

L32: “The gaze of the predators and prey”

INTRODUCTION

- Because both human and non-human studies are addressed, it is important to mention this more explicitly. It is currently not always clear (e.g. L52)

- From the first paragraph in the introduction it should be clear to the reader that attention is a limited cognitive capacity, and that this explains why attention is selected for relevant stimuli, e.g. animate objects.

- Throughout the manuscript it is not really discussed that the prey is not necessarily a neutral animal, as (ancestral) humans hunt(ed) these species, making humans both a predator and a prey. Similar arguments can be made for chimpanzees.

- Within the introduction there is a mix between eye contact with conspecifics and heterospecifics for which we can expect different attentional and motivational systems. Barrett 2015 has written a chapter that explains this nicely: Barrett, H. C. (2015). Adaptations to predators and prey. The handbook of evolutionary psychology, 200-223. This chapter can be useful in general as they touch on the topic of forward-facing eyes of predators (p 208).

L67: You could strengthen your argument here by mentioning that it is likely that for most participants in these studies these are innate responses as they probably have never encountered such animals in real life (unless you have information about the exposure of the humans to lions).

L68-71: I think this sentence can be removed as it is focusing on conspecific interactions, which are fundamentally different from predator-prey interactions.

L73: I think there is a number of primate papers that are missing that would strengthen this argument, especially since you are comparing human and chimpanzees.

The following paper found evidence for attention biases in bonobos for leopards:

Laméris, D. W., Verspeek, J., Eens, M., & Stevens, J. M. G. (2022). Social and nonsocial stimuli alter the performance of bonobos during a pictorial emotional Stroop task. American Journal of Primatology, 84(2), e23356. https://doi.org/10.1002/ajp.23356

The following papers report attention biases for poisonous animals (e.g., snakes)

Masataka, N., Koda, H., Atsumi, T., Satoh, M., & Lipp, O. V. (2018).

Preferential attentional engagement drives attentional bias to snakes in Japanese macaques (Macaca fuscata) and humans (Homo sapiens). Scientific Reports, 8(1), 17773. https://doi.org/10.1038/s41598-018-36108-6

Hopper, L.M., Allritz, M., Egelkamp, C. L., Huskisson, S.M., Jacobson, S. L., Leinwand, J. G., & Ross, S. R. (2021). A comparative perspective on three primate species’ responses to a pictorial emotional stroop task. Animals, 11(3), 588. https://doi.org/10.3390/ani11030588

Shibasaki, M., & Kawai, N. (2009). Rapid detection of snakes by japanese monkeys (Macaca fuscata): an evolutionarily predisposed visual system. Journal of Comparative Psychology, 123(2), 131–135. https://doi.org/10.1037/a0015095

L77: Replace 'new' with 'recent'

L87-89: I feel Kano et al. (2015) is missing here. They report that bonobos, as compared to chimpanzees, pay more attention to eye regions.

Kano, F., Hirata, S., & Call, J. (2015). Social attention in the two species of pan: Bonobos make more eye contact than chimpanzees. PLoS ONE, 10(6). https://doi.org/10.1371/journal.pone.0129684

L89-91: Please highlight that this is likely more true between conspecifics, whereas the current paper is focusing on heterospecifics

L92: Please see Tomonaga & Imura (2015) DOI: 10.1038/srep11437

L97-100: I think it's debated whether the attention bias for snakes/spiders is due to their face. For example, in snakes the shape and presence of the scales drive responses (https://doi.org/10.1111/psyp.12564 and Kawai, N., & Kawai, N. (2019). Searching for the Critical Features of Snakes. The Fear of Snakes: Evolutionary and Psychobiological Perspectives on Our Innate Fear, 121-153.) Please rephrase this sentence.

L137: "or other alternative hypotheses" This is unclear, what do you exactly mean?

L138: Please elaborate on other reasons why chimpanzees would not show similar attentional biases? There is literature on pareidolia effects in primates that can be useful to discuss whether or not you would expect similar results.

METHODS

- Eye-tracking Analysis: I understand that for ape studies the trial duration is preferably shorter than for human studies. However, for a more fair comparison of the two species I would think you can make the argument to only analyse the first 3s of the human data. An extension of 5s is significantly long to capture additional attentional processes. I realise this might be a lot to request, and perhaps I am missing the rationale of why you looked at the full 8s human trials, but as you are comparing humans with chimpanzees I would expect a more similar study design.

L158: Please report whether students provided consent

L174: It is not fully clear if both the Free-view task and Danger Rating task consisted of 96 images. Please clarify. Did the two tasks use the same images?

I see now that you later discuss this at L199. Perhaps to avoid confusion you can mention this earlier?

L202-204: But did the human participants rate the lions are more dangerous than the impalas?

L212-214: Is the original range of the two wild-born chimpanzees known? Where they actually savanna dwelling chimpanzees?

L264-267: If a first trial was deemed successful, were subsequent repeated trials still included in the analysis? There could be a risk of habituation effects. This could be something you can additionally test to ensure whether or not this occurred.

L281: Please highlight that you used pictures of female impalas, as earlier mentioned in L149

L281: I appreciate the author's efforts into finding this large number of different stimuli as it can be challenging to collect sufficient suitable stimuli. I think it would increase transparency if you mention whether you aimed to control for background features, or body orientation of the animals. E.g. for directed images the animal can be approaching the camera, facing the viewer, or can walk parallel to the camera but turn its head towards the camera (e.g. the two impala images in Figure 1). I'm aware that you partially corrected for this difference in ROI area in the processing, but it would benefit the reader if this information is provided.

L300: As you are analysing initial fixations it is really important that you report how the fixation cross proceeded. For the human study this is more controlled as the participants decided themselves when they continued, but for the chimpanzees this is not clearly reported. Was this manually done, after an Xms duration or when the chimpanzee was fixating at the cross. Eye-tracker systems vary in how this can be achieved. Additionally, did you verify whether the participants were in fact focusing on the fixation cross?

Statistical analysis: I would like to ask for the rational to go for ANOVAs over mixed models. Did the data meet the assumptions for an ANOVA (homogeneity of variance, normality). I was under the impression that ANOVAs are typically not suitable for proportional data, especially when there are a lot of extreme values (0 or 100 in this case). Wouldn't a mixed model with an appropriate distribution be better? This way you would be able to control for number of trial repetitions for example.

L328: Please mention what these a priori contrasts were

RESULTS

- In my opinion, the Results section would improve by clearly mentioning the model which is being discussed. So either, label the different models (model 1, model 2, model 3) in the "statistical analysis" and present the results in this way. Or perhaps better, present the results in terms of the predictions. Because you use one model for multiple predictions, this can create some confusion of what is actually being discussed.

- My understanding is that in a significant three-way interaction, looking at significant two-way interaction effects can be misleading, or incomplete, as the three-way interaction suggests that the two-way interaction depends on the third variable. Given that you report a significant three-way interaction for both humans and chimpanzees, I'm not fully convinced that the two-way interactions you report are meaningful. I'm aware that looking at two-way interactions can be meaningful in certain cases, but if you think this applies to your results I think it would be good practice to explicitly mention this. For example, you report that there is a difference in fixation proportion for the impala and lion eyes (L355-356), however, when looking at Figure 2 there appears a modulating effect of the gaze direction for the lion stimuli that possibly explains this effect. You discuss this in the lines after, but in reporting your results you jump between mentioning the three-way and two-way effects which is confusing. Given the presence of the three-way interaction I think it is more appropriate to report the significant effects and then focus on the insignificant contrasts within that three-way interaction, rather than the significant two-way interactions.

- Is there a reason why you did not directly test if the (initial) fixation proportion was associated with the saliency of the ROI? To me this would provide more reliable evidence that gaze patterns were not influenced by such low-level features.

DISCUSSION

- Throughout the discussion I'm missing discussion that humans have arguably different relationships with lions/impalas than (captive) chimpanzees. For example, lions are typically considered charismatic species that are highly favoured to see in zoos or safaris. This is not to say that this necessarily drove your results (especially as you found very similar results between humans and chimpanzees), but I would like to see more discussion on how the difference in results between the species can be explained

- I'm missing a more detailed discussion into why the chimpanzees seemed to fixated more on the impala bodies. Could it be that you find similar results in humans when you focus on the first 3 seconds?

L415: Please highlight again that this partly supports the first prediction (which included both humans and chimpanzees) as you could only test this for the chimpanzees. Additionally, perhaps you can make suggestions as how future studies can test this in a better way (i.e., avoiding ceiling effects as found with the human participants). E.g., showing impala and leopard pictures at the same time.

L422-424: Please highlight that gaze patterns for conspecifics can be different from those for heterospecifics. Guo et al. 2019 (https://doi.org/10.1007/s00221-019-05574-3) comes to mind that reports some work on this.

L431: Please highlight that this is mostly for nonhuman heterospecific animals as there are multiple studies that report this when using conspecific stimuli.

L433: For example, this statement is based on the two-way interaction whereas Figure 2 suggests a three-way interaction. Here you would need to report if the humans and chimpanzees also fixated more on the averted lion eyes than the averted impala eyes.

L445-451: This paragraph is not very convincing in explaining why the saliency analyses show that bottom-up processing does not account for the gaze patterns. If this is mostly based on the fact that the differences in saliency do not align with the differences in gaze patterns, I do not think this is sufficient to exclude bottom-up processing

L457-459: Kano 2009 (doi:10.1098/rspb.2008.1811) and Kano 2011 (DOI 10.1007/s10071-011-0422-5) can be useful here where they report that chimpanzees after faster fixation rates than humans

L459: Please change 'is' to 'could be'

L466-468: This is just a little confusing because you look at both 3- and 2-way interactions which result in somewhat contrasting results

REFERENCES

- Kano et al. 2018 (L429) is missing from the reference list

- Please add the year in Yorzinski L669

6. PLOS authors have the option to publish the peer review history of their article (what does this mean?). If published, this will include your full peer review and any attached files.

Reviewer #1: No

Reviewer #2: No

---

## [Author Response · Author response to Decision Letter 0]

11 Apr 2024

Note that our complete response to reviewers is both included below and uploaded as a Word document in our submission materials.

Reviewer #1 comments

R1C1 :: This is an interesting and well-written manuscript about a topic that has not attracted much prior research and as such is welcomed. There were however some important deficiencies in theoretical justification and experimental design (or insufficient explanation) that need to be addressed. Perhaps the authors have already thought through and solved many of these, but the reasoning should be included in the manuscript for readers to follow all points of their logic. 

Response :: We thank the reviewer for their constructive review of the manuscript and for giving us the opportunity to present a more clearly reasoned, easier-to-follow manuscript. 

Major concerns: 

R1C2 :: The paper refers throughout to predator and prey gaze, but tests only lion and impala with no other species. This seriously limits the generalizability of their conclusions. This is compounded by fairly weak justification of species choice. It is true lions co-occur with eastern chimpanzees in some narrow savannah regions where chimpanzees are likely hunted to at least some extent, but most captive chimpanzees, and presumably the current population, are western chimpanzees. Beyond subspecies differences, most chimpanzees live in forest environments. For both of these reasons, leopards are much more ecologically relevant and indeed have been found to predate chimpanzees across their range more than has been reported for lions. Further, the stimuli examples appeared to show male lions, who do not typically engage in much of the hunting, and lions further are group-hunters (not solitary as are leopards) and only one individual at a time was presented. While the big cat morphology may be a form that is relatively conserved, and predates upon many primates across the world including humans and chimpanzees, the text only makes a very vague association with one species in one part of a different subspecies’ range. I recommend justifying this point in the text, that we may expect conserved attentional biases to come through across big cat morphologies including the lions used in this study. Relatedly, on the other side, chimpanzees are proficient hunters themselves but do not hunt impala. Duikers, squirrels, red colobus monkeys, would all be more species-relevant prey to a chimpanzee, why was impala chosen? This should also be mentioned when discussing why human attention may differ from other species. Chimpanzees, like humans, are both predators are prey to different animals, but the stimuli only included species that are predator and prey to humans. It is thus not really a fair species comparison. The discussion certainly should mention the lack of species diversity as a limitation on generalizability and future direction, even moreso in light of these larger concerns. 

Response :: We agree that the language of our original submission both obfuscated our intent and suggested a set of ecological/evolutionary inquiries that our experiments could not effectively test. We have clarified throughout (ll 140-164; 165-171) that our questions, hypotheses, and experiments were targeted to gaze behaviors related to the prototypical features of predators (e.g., forward-facing eyes, robust body) and prey (e.g., lateral eyes) exemplified by the lion and impala. We make no claims about primate behavioral ecology except as related to any deeply conserved attentional biases that ancestral primate evolutionary history may have introduced. The lack of diversity in our sample of predator and prey images is indeed a sincere limitation (albeit an intentional one, designed to measure attentional biases in limited time with a highly unique chimpanzee population) and an appropriate target for future studies. We have added language about this limitation to the discussion (ll 474-478). 

R1C3 :: The inclusion of the saliency maps felt a little out of place and I did not find them integrated into the rest of the discussion. Some introductions of their actual use and predictive power in the introduction would be of use. As of now, they are introduced only in ways they fail to predict gaze, and then the results again fail to find them making new predictions, and the naïve reader is left wondering what these can contribute if they consistently fail to make predictions. 

Response :: We have added a paragraph to the introduction (ll 181-194) to justify the inclusion of salience maps to test the conservative (and arguably default) alternative hypothesis that any differences in gaze behavior can be understood, with fewer assumptions, to be the result of low-level stimulus features. We believe that such tests are a prerequisite for making more complex arguments from cognitive representation and primate evolutionary biology that our predictions require. 

R1C4 :: The hypotheses are quite vague and not well-justified, which combined with a very high number of statistical tests leads readers to wonder if this was more exploratory or hypothesis-driven. Both are welcomed, especially with new topics such as this, but perhaps the manuscript could benefit from distinguishing them and making clearer, concrete predictions associated with the actual hypotheses (ideally with citation). In particular, the first set of predictions (paragraph starting line 131) is essentially just sameness or difference, but then no statistical tests actually directly compared the two species. The lack of direct human-chimpanzee comparison is a strange choice. The reasoning of the sentence starting on line 135 is especially vague and nebulous. If these were the topics of interest wouldn’t a study comparing humans with different background knowledge and social contexts be a more salient test? The choice of a cross-species framework here could be much better-justified. 

Response :: We acknowledge that the original formulation of our predictions was not always clear, and we have substantially edited these sections referenced above to better convey the intent of our experiments (ll 140-164). We tested the same set of predictions – about eye direction, prototypical predator and prey features, etc – in both humans and chimpanzees with an eye towards demonstrating which primate attentional biases may have been maintained over evolutionary time. We have clarified that we do not expect differences among the humans and chimpanzees’ gaze behaviors, as these would suggest some outsized influence of background knowledge, sociocultural forces, learning, or the other enculturating forces Reviewer 2 identified (“humans have arguably different relationships with lions/impalas than (captive) chimpanzees. For example, lions are typically considered charismatic species that are highly favoured to see in zoos or safaris.”), which we did not predict nor intend to test (ll 140-152). The nature of these human-chimpanzee hypotheses is a poor fit for statistical tests since the parsimonious, literature-informed hypothesis is the null, which we cannot test, since our predictions are primarily about which species/regions are fixated most frequently (not their absolute proportion or rank order), and since any effects are nested in complex interactions with all of the other factors we tested. 

R1C5 :: The dependent measure as proportion of fixation is very unusual compared to previous great ape work. Why was total fixation duration not included? What reasoning is there for choosing number of fixations over duration, especially considering prior work consistently showing chimpanzee gaze is likely to jump around more than humans. Could the lion have evoked more glances away (chimpanzees tend to avoid prolonged direct eye contact) while the averted gaze and impala had longer but more consistent gaze? At the very least, supplemental analysis detailing the duration should be included somewhere. 

Response :: Our dataset is indeed well-suited to many analytical frameworks. Our choice of dependent measure, over other correlated measures, is aligned with the human work published using area-normalized fixation proportions (Birmingham et al., 2008a, 2008b). We acknowledge that fixation duration and fixation number are slightly different measures, but generally strongly correlated. No new effects emerge from our dataset when the dependent measure is changed to area normalized total fixation duration, now featured as a supplement figure 4. 

Minor concerns: 

R1C6 :: Line 87: The topic of chimpanzees seems to come out of nowhere here. As they are central to the whole paper, their introduction should be naturally connected. Perhaps a linking phrase of paragraph about the value and perspective around comparative gaze studies of chimpanzees would help improve this and address one of the major comments above? 

Response :: The structure and language of the introduction, especially with regard to the species under study, has been updated for clarity throughout. Several additional primate eye-tracking designs have been added or elaborated upon in this revision (ll 56-57; 66-73; 91; 93-95). 

R1C7 :: Line 87: This finding itself also needs more rigour if chimpanzee attention to faces is a central focus of the paper. The previous studies find that chimpanzees gaze more at mouth than eyes even within faces, and this has been considered as potentially more species-relevant for social cues than eyes (Kano et al. 2018, Brooks et al 2021). Did the authors consider looking at attention to mouth by species given this finding? At the very least, the difference in the role and meaning of eye-contact across species should be mentioned. Note also that the supplemental material of some of these previous face-based studies have looked at attention to direct compared to averted gaze, which while not directly on the topic of predator/prey gaze may be useful for the authors to look at. 

Response :: We have added language on chimpanzees’ relative interest in other face regions besides eyes (ll 90-92). As emphasized by reviewer 2, we are sensitive to the idea that mechanisms of social gaze to conspecifics are distinct from mechanisms of gaze to potentially dangerous heterospecifics, with the latter the target of our research with humans and chimpanzees. 

R1C8 :: Line 116: This is a bit of a strange citation here, as the point of the sentence is that eyes have high saliency in the face but the cited paper is on the topic of that not being at all universal, and certainly is not a citation about them being “presented in this manner.” 

Response :: We have clarified our use of this citation. 

R1C9 :: Line 139: This whole paragraph does not distinguish species for the predictions. Maybe the authors intend these predictions to apply to all participants, but this should then be made explicit. 

Response :: As above, we have substantially edited the way we present our predictions to clarify what we were (and were not) testing (ll 141-169). 

R1C10 :: Line 161: The ethnicity states Chinese, Korean, and east Asian as distinct, please check this or clarify. 

Response :: This was the language of the demographic survey. 

R1C11 :: Throughout: “other part of the head” may be clearer than just head when distinguishing from eyes, since in some sentences this could easily be lost by readers. 

Response :: To avoid confusion we have language in the methods pointing the reader to Figure 1B for a visualization of what is meant by eyes, head, body, and background. 

R1C12 :: Several times: It may be a bit inaccurate to say we couldn’t look at lion-impala differences in humans because of high attention to both. It would be more accurate to say no differences were observed because of ceiling effects, or the data was not suited for a formal model because of ceiling effects but visually appeared identical. Saying both were at ceiling is a comparison itself. 

Response :: We thank you for this statistical point, and have corrected our language (ll 405-406). 

R1C13 :: Line 430: “To our knowledge, this experiment provides some of the first evidence examining the strength of the bias towards visually fixating the eyes of nonhuman animals in both human and chimpanzee participants.” This sentence must be rephrased or removed. The prior sentence cites Kano et al. 2018, which compared several primate species including humans on social attention. Also, the nonhuman animals point is somewhat missed here if talking about nonhuman participants, maybe allospecific would be more relevant. 

Response :: We have rephrased this point to make clearer that what we find novel is the measured strength of bias to lion and impala eyes, rather than the kinds of within-species / within-primate comparisons that are already part of the cited literature (ll 492-494). 

R1C14 :: General: the size of the eye ROIs appears very small in the figure, especially considering the screen size used. Are these within the margin of error of accuracy? This should be mentioned. 

Response :: We have now noted that all the ROIs were within the error accuracy of the trackers (ll 352-353). 

R1C15 :: Line 443: Chimpanzees are one of the nearest relatives, bonobos are equally closely related. Bonobos generally look much more to eyes than chimpanzees (Kano et al. 2018) and have no lions in their range (but do also face leopard predation) and are less proficient hunters. Perhaps a future comparative approach can be highlighted as a future direction? 

Response :: We have added this suggestion for a target of future research (ll 531-534). 

R1C16 :: Line 444: check divergence dates. 

Response :: We have removed this extraneous language. 

R1C17 :: General: the discussion could use more interpretation and connection to the broader literature, as it stands it is hard to see what the importance beyond restatement of the results. 

Response :: We have revised the Discussion to better highlight what is novel about our results (ll 516-517), how our results were unlikely to be the product of forces other than evolved attention biases (ll 509-511; 518-520), and how future designs could further explore this topic (ll 474-478; 531-534). 

R1C18 :: One point not raised anywhere in the manuscript that certainly bears discussion: predators have more forward-facing eyes than prey. Could you rule out the fact that there is a general direct eye-gaze bias, but that the lions just have more direct gaze (at least from the view of primates such as ourselves with forward-facing eyes) and it is therefore not related to predator/prey distinctions per se? 

Response :: This is an important point, and indeed something we are very interested in. We are much more explicit in the revision that exactly these differences – between a prototypical predator phenotype and a prototypical prey phenotype – are likely determinants of differences in patterns of gaze behavior across images (ll 140-142; 154-157). Future research could clarify what features of our prototypical predator and prey are most responsible for gaze behavior biases in human and nonhuman primate attention to a much more diverse set of stimuli (ll 474-478). 

R1C19 :: How do you account for the fact that the relative eye size between the two species is different? The eyes of the impala appear to be a much bigger proportion of the head than lions. Perhaps this strengthens the finding, but it bears discussion. 

Response :: The area-normalized fixation proportions that we used as the dependent measure in all analyses scale fixation counts to a region-of-interest by the proportion of the screen that the region-of-interest occupies. The relative size of the eye region was not necessarily larger for either or lion images or our impala images. 

R1C20 :: It is rare in eye-tracking with great apes to use a mesh rather than transparent barrier between eyes and screens. Can you be sure this did not affect gaze tracking when one or the other eye is blocked to the sensor? 

Response :: The mesh was an unfortunate requirement of our access to the chimpanzees. We are nevertheless confident in our

---

## [Decision Letter · Decision Letter 1]

7 May 2024

PONE-D-23-40092R1Predator gaze captures both human and chimpanzee attentionPLOS ONE

Dear Dr. Whitham,

Thank you for submitting your manuscript to PLOS ONE. After careful consideration, we feel that it has merit but does not fully meet PLOS ONE’s publication criteria as it currently stands. Therefore, we invite you to submit a revised version of the manuscript that addresses the points raised during the review process.

We look forward to receiving your revised manuscript.

Kind regards,

Nick Fogt

Academic Editor

PLOS ONE

Additional Editor Comments:

Thank you for submitting your revised manuscript.

One of the original reviewers has reviewed the revised paper. The reviewer continues to have major concerns regarding the implication in the paper that those data from the manuscript allow for a comparison between human and chimpanzee behavior. Please address to what extent such comparisons can (or cannot) be made, particularly given that the experimental procedures vary. The reviewer has offered a number of suggestions that if followed, may help to address these concerns.

Reviewers' comments:

Reviewer's Responses to Questions

**Comments to the Author**

1. If the authors have adequately addressed your comments raised in a previous round of review and you feel that this manuscript is now acceptable for publication, you may indicate that here to bypass the “Comments to the Author” section, enter your conflict of interest statement in the “Confidential to Editor” section, and submit your "Accept" recommendation.

Reviewer #2: (No Response)

2. Is the manuscript technically sound, and do the data support the conclusions?

Reviewer #2: No

3. Has the statistical analysis been performed appropriately and rigorously? 

Reviewer #2: No

4. Have the authors made all data underlying the findings in their manuscript fully available?

Reviewer #2: Yes

5. Is the manuscript presented in an intelligible fashion and written in standard English?

Reviewer #2: Yes

6. Review Comments to the Author

Reviewer #2: I would like to thank the authors for responding to my previous comments. I appreciate some of the adjustments made by the authors which already improved the quality of the manuscript. However, there remain several points where I believe the authors' responses were not entirely adequate which I list below.

MAJOR COMMENTS

1. I continue to encounter major concerns with the difference in study designs for the human and chimpanzee study. While I acknowledge the authors’ clarification that it was not the intention to formally test species differences, many sentences throughout the manuscript suggest a comparability between the human and chimpanzee findings:

L36-38: “The striking similarities between the gaze patterns of humans and chimpanzees provide additional evidence that attentional processing of two-facing eyes is evolutionarily conserved across primates.”.

L141-144: “Attentional biases for predators over prey in modern humans were likely shaped by the same sources of natural selection as those experienced by chimpanzees and as a result patterns of chimpanzee attention should reflect the same (or similar attentional biases)”. You need to statistically test for this. Simply comparing patterns is not scientifically sound.

L486-488: “These chimpanzee results cohere with prior research suggesting that apes attend to eyes of conspecifics and heterospecifics but do so less often than human participants (Kano et al., 2018).”

L514-530: This entire paragraph compares the chimpanzee results to the human results. I don’t see how this is justified if you used a different study design and therefore potentially investigated two completely different things.

2. Linked to my previous point, the authors’ response fails to address why a similar study design wasn’t initially employed, or why they do not want to analyze only the first 3s of the human data, as previously suggested. The authors replied stating that chimpanzees saw three repetitions of each image, but the text mentions that each image was shown four times. Moreover, the argument that by repeating the trials was intended to roughly control the total screen time does not address my concern. The discrepancy between eight consecutive seconds and 3 sets of 3 seconds each is substantial, potentially leading to disparate gaze patterns and undermining the comparability of human and chimpanzee trials (e.g., it is unlikely that the chimpanzees would pick up their gaze patterns where they left in the previous trial). I strongly recommend the authors to reconsider only analyzing the first 3s of the human trials so that a comparison can actually be made. I am aware that the authors state that a human-chimpanzee comparison was not a study aim, but above I mention a couple of places in the paper where this is not clear and where readers can easily misinterpret this. In this way, the title and abstract are already not transparent as there is no mention of this difference in study design. Not even in the discussion is this limitation of the study mentioned.. Similarly for the predictions, I appreciate that the authors included the sentence stating that they did not make specific predictions about divergent patterns, but you actually ran two experiments with different study designs. This needs to be mentioned and you need to make specific predictions about this. Alternatively, the manuscript needs to be restructured in a way that it becomes clear that you tested the humans and chimpanzees on different tasks.

3. I thank the authors for their additional explanation regarding their statistical approach. I'm, however, still not convinced about including a three-way interaction but focusing on the significant two-way interactions (because these were planned contrasts). Such a selective approach may be perceived as cherry-picking, as it disregards other potential effects that may exist within the data. It also raises the question why, for example, the comparison between lion*eye*direct versus lion*body*direct is not made. From the graph it suggests that this effect is there and, to me, would give much stronger evidence for an attention bias for the eyes of a predator, incorporating your reasoning that the direction of the face is an important modulating factor. By ignoring the significant three-way interaction you're missing out on reporting results.

In L412 for example, you find a significant three-way interaction, indicating that the fixation proportion is influenced by an interplay between those three factors. But here, by looking at the two-way interaction your possibly just ignoring a possible significant effect between lion*head*direct and lion*body*direct. Alternatively, in the next sentence you focus on the significant stimulus type*ROI effect and report that humans and chimpanzees fixated more on lion eyes than the impala eyes, but from the graph it is obvious that gaze direction modulates this effect (for the lions).

4. Additionally, I find the authors' response as to why the saliency was not directly tested against the (initial) fixation proportion unsatisfactory. Mentioning that the data will be made available for alternative approaches that are outside the scope of your questions is not true, and truthfully suggests reluctance from the authors to consider changes. It’s essential to note that the current methodology precludes conclusive remarks on the impact of saliency on gaze patterns, as highlighted in specific sections of the manuscript (e.g., L504-508, L532—535). Given that the effect of saliency on AOIs is one of your main predictions, it is essential to use proper statistical methods to actually make any conclusions. I very much appreciate the inclusion of the saliency maps in this paper which is exemplary for other studies, but if the authors do not want to properly test this, I would suggest removing this component from the study. I reiterate the recommendation for using mixed models which would allow for incorporating the saliency values as a control variable or covariate.

5. I agree with Reviewer 1 that within the primate literature total fixation duration is more common. While the authors reply that proportion of fixations aligns with previous human work, this could similarly be applied to primate studies, so it is not clear why this approach was favored over the other. I appreciate the addition of figure S4 showing these data, and the authors state that no new effects emerged from the datasets with normalized total fixation duration is used as dependent variable. However, no statistics are provided or indicated in the figure. Again, simply comparing patterns between graph is not sufficient to make such conclusions. Even then, just by scanning figure S4 and comparing with figure 2B, it seems clear that the two do not follow the same pattern (but then again, we would need statistics to know this, just like with the saliency results). It is also unclear why only chimpanzee data are shown and not human data.

MINOR COMMENTS

L68-69: I would add the species for each of the references.

L71: Replace ‘lions’ with 'animals' or something alike as you don't only mention lions.

L73-75: I would move this to the next paragraph where you actually introduce that faces are relevant.

L95: You start the sentence with “One study found…” but you actually refer to multiple studies, please revise.

L115: I would add the argument of the pareidolia effect at the end of this sentence to highlight that this is such a strong tendency that it extends beyond natural faces.

L122: Senju et al. 2011 is an interesting paper here: https://doi.org/10.1080/13506280444000157

L182: Rephrase: “and the face being directed or averted”

L376: “Section 3.1.1”

L388: "stimulus species", or something to highlight it's about the lion and impala.

L410: If you used a linear mixed-effects model, I assume you included random effects. What were these?

Table 2: Having a second look at the table, the comparison Background direct vs. Background averted is missing.

L475-479: But you did not test the difference between for example impala head and body, so you cannot conclude this (or these statistics are at least not reported in the text and tables). At the same time, this was tested for in the saliency analyses. This raises concerns regarding the clarity and coherence of the testing procedures, potentially impeding readers' understanding of the tested variables and their corresponding results. Given this inconsistency, readers may find it challenging to track and interpret the outcomes of the various analyses conducted.

L484-486: Again, not listed in the tables.

7. PLOS authors have the option to publish the peer review history of their article (what does this mean?). If published, this will include your full peer review and any attached files.

Reviewer #2: No

---

## [Author Response · Author response to Decision Letter 1]

16 Aug 2024

Reviewer comments________________________________________

I would like to thank the authors for responding to my previous comments. I appreciate some of the adjustments made by the authors which already improved the quality of the manuscript. However, there remain several points where I believe the authors' responses were not entirely adequate which I list below.

MAJOR COMMENTS

RC1 :: I continue to encounter major concerns with the difference in study designs for the human and chimpanzee study. While I acknowledge the authors’ clarification that it was not the intention to formally test species differences, many sentences throughout the manuscript suggest a comparability between the human and chimpanzee findings:

L36-38: “The striking similarities between the gaze patterns of humans and chimpanzees provide additional evidence that attentional processing of two-facing eyes is evolutionarily conserved across primates.”.

L141-144: “Attentional biases for predators over prey in modern humans were likely shaped by the same sources of natural selection as those experienced by chimpanzees and as a result patterns of chimpanzee attention should reflect the same (or similar attentional biases)”. You need to statistically test for this. Simply comparing patterns is not scientifically sound.

L486-488: “These chimpanzee results cohere with prior research suggesting that apes attend to eyes of conspecifics and heterospecifics but do so less often than human participants (Kano et al., 2018).”

L514-530: This entire paragraph compares the chimpanzee results to the human results. I don’t see how this is justified if you used a different study design and therefore potentially investigated two completely different things.

Response :: We have removed the specific sentences of major concern. We have also, in response to several of the constructive comments below, substantially restructured the manuscript to more completely disambiguate the two experiments given the differences in design and methodology. 

RC2 :: Linked to my previous point, the authors’ response fails to address why a similar study design wasn’t initially employed, or why they do not want to analyze only the first 3s of the human data, as previously suggested. The authors replied stating that chimpanzees saw three repetitions of each image, but the text mentions that each image was shown four times. Moreover, the argument that by repeating the trials was intended to roughly control the total screen time does not address my concern. The discrepancy between eight consecutive seconds and 3 sets of 3 seconds each is substantial, potentially leading to disparate gaze patterns and undermining the comparability of human and chimpanzee trials (e.g., it is unlikely that the chimpanzees would pick up their gaze patterns where they left in the previous trial). I strongly recommend the authors to reconsider only analyzing the first 3s of the human trials so that a comparison can actually be made. I am aware that the authors state that a human-chimpanzee comparison was not a study aim, but above I mention a couple of places in the paper where this is not clear and where readers can easily misinterpret this. In this way, the title and abstract are already not transparent as there is no mention of this difference in study design. Not even in the discussion is this limitation of the study mentioned.. Similarly for the predictions, I appreciate that the authors included the sentence stating that they did not make specific predictions about divergent patterns, but you actually ran two experiments with different study designs. This needs to be mentioned and you need to make specific predictions about this. Alternatively, the manuscript needs to be restructured in a way that it becomes clear that you tested the humans and chimpanzees on different tasks.

Response :: We have followed the advice to restructure the manuscript in a way that more clearly emphasizes the differences of procedure, analyses, and conclusions between the human and chimpanzee designs. In the general discussion we again emphasize these differences alongside the conservative inference that in both humans and chimpanzees we observed evidence of the biases to lions and to their directed gaze that we predicted (ln 575-590). We intended 3 repetitions of a stimulus to indicate the four total exposures described in the manuscript, but apologize for the unclear language.

RC3 :: I thank the authors for their additional explanation regarding their statistical approach. I'm, however, still not convinced about including a three-way interaction but focusing on the significant two-way interactions (because these were planned contrasts). Such a selective approach may be perceived as cherry-picking, as it disregards other potential effects that may exist within the data. It also raises the question why, for example, the comparison between lion*eye*direct versus lion*body*direct is not made. From the graph it suggests that this effect is there and, to me, would give much stronger evidence for an attention bias for the eyes of a predator, incorporating your reasoning that the direction of the face is an important modulating factor. By ignoring the significant three-way interaction you're missing out on reporting results.

In L412 for example, you find a significant three-way interaction, indicating that the fixation proportion is influenced by an interplay between those three factors. But here, by looking at the two-way interaction your possibly just ignoring a possible significant effect between lion*head*direct and lion*body*direct. Alternatively, in the next sentence you focus on the significant stimulus type*ROI effect and report that humans and chimpanzees fixated more on lion eyes than the impala eyes, but from the graph it is obvious that gaze direction modulates this effect (for the lions).

Response :: We appreciate the additional notes on statistical interpretation of the results. The full matrix of three-way post hoc comparisons are briefly discussed in new sections added to the results for Experiment 1 (ll 358-370) and Experiment 2 (ll 539-545), visualized in new supplements S1 and S7, and reported explicitly in supplements S2 and S8. We attempted throughout to tailor our analyses to the specific predictions we made about anticipated patterns of gaze data. Planned contrasts were targeted to our specific statistical predictions about anticipated differences that may emerge from the hypothesized attentional biases that are the subject of the manuscript. 

RC4 :: Additionally, I find the authors' response as to why the saliency was not directly tested against the (initial) fixation proportion unsatisfactory. Mentioning that the data will be made available for alternative approaches that are outside the scope of your questions is not true, and truthfully suggests reluctance from the authors to consider changes. It’s essential to note that the current methodology precludes conclusive remarks on the impact of saliency on gaze patterns, as highlighted in specific sections of the manuscript (e.g., L504-508, L532—535). Given that the effect of saliency on AOIs is one of your main predictions, it is essential to use proper statistical methods to actually make any conclusions. I very much appreciate the inclusion of the saliency maps in this paper which is exemplary for other studies, but if the authors do not want to properly test this, I would suggest removing this component from the study. I reiterate the recommendation for using mixed models which would allow for incorporating the saliency values as a control variable or covariate.

Response :: The salience analyses have been reworked to more directly test the predictive validity of salience values on initial fixation proportions using a framework that tests the salience at the fixated region to two chance-based estimates (ll 280-296; 347-356; 533-537). Salience maps do not predict the pattern of gaze behaviors that were the subject of our design, analyses, and conclusions. 

RC5 :: I agree with Reviewer 1 that within the primate literature total fixation duration is more common. While the authors reply that proportion of fixations aligns with previous human work, this could similarly be applied to primate studies, so it is not clear why this approach was favored over the other. I appreciate the addition of figure S4 showing these data, and the authors state that no new effects emerged from the datasets with normalized total fixation duration is used as dependent variable. However, no statistics are provided or indicated in the figure. Again, simply comparing patterns between graph is not sufficient to make such conclusions. Even then, just by scanning figure S4 and comparing with figure 2B, it seems clear that the two do not follow the same pattern (but then again, we would need statistics to know this, just like with the saliency results). It is also unclear why only chimpanzee data are shown and not human data.

Response :: We clarify the relative merits of using area-normalized fixation proportions for designs of this kind in the Eye-tracking analysis text (ln 264-272). We have added ANOVA and planned comparisons analyses on total fixation duration to figure S4 to parallel those of the main analysis of the manuscript. Parallel analyses with the human data are unfortunately not possible using as the software did not record fixation times to background or off-screen fixations. As with the planned contrasts above, we favor a selective approach that is targeted to the specific predictions we made about anticipated patterns of gaze data using the dependent variable we intended to base our analyses on. 

MINOR COMMENTS

RC6 :: L68-69: I would add the species for each of the references.

Response :: We have made this change (ln 63-67)

RC7 :: L71: Replace ‘lions’ with 'animals' or something alike as you don't only mention lions.

Response :: We have made this change (ln 67-70)

RC8 :: L73-75: I would move this to the next paragraph where you actually introduce that faces are relevant.

Response :: We have made this change (ln 79-81)

RC9 :: L95: You start the sentence with “One study found…” but you actually refer to multiple studies, please revise.

Response :: We have made this change (ln 94-96)

RC10 :: L115: I would add the argument of the pareidolia effect at the end of this sentence to highlight that this is such a strong tendency that it extends beyond natural faces.

Response :: We have made this change (ln 112-114)

RC11 :: L122: Senju et al. 2011 is an interesting paper here: https://doi.org/10.1080/13506280444000157

Response :: One of our favorites, cited ln 95-96

RC12 :: L182: Rephrase: “and the face being directed or averted”

Response :: We have made this change (ln 172-174)

RC13 :: L376: “Section 3.1.1”

Response :: We have added “Section” to each leading reference throughout the manuscript.

RC14 :: L388: "stimulus species", or something to highlight it's about the lion and impala.

Response :: We have changed stimulus type to “stimulus species” throughout the manuscript

RC15 :: L410: If you used a linear mixed-effects model, I assume you included random effects. What were these?

Response :: These salience models are no longer a part of the manuscript in favor of the new approach to salience analyses described above.

RC16 :: Table 2: Having a second look at the table, the comparison Background direct vs. Background averted is missing.

Response :: The area-normalized fixation proportions are normalized such that the proportions sum to one, allowing for only 1 unique comparison in a comparison of only animals vs backgrounds. This aside, Table 2 is omitted from the rearranged manuscript since only the chimpanzee data were interpretable.

RC17 :: L475-479: But you did not test the difference between for example impala head and body, so you cannot conclude this (or these statistics are at least not reported in the text and tables). At the same time, this was tested for in the saliency analyses. This raises concerns regarding the clarity and coherence of the testing procedures, potentially impeding readers' understanding of the tested variables and their corresponding results. Given this inconsistency, readers may find it challenging to track and interpret the outcomes of the various analyses conducted.

Response :: Table 4 is incidentally not needed in the revised manuscript due to the reconfiguration of salience analyses to use Mann-Whitney tests to more directly test the relationship between initial fixations and salience. The tests targeted to our specific hypotheses should be consistent throughout the manuscript, and all unplanned comparisons are available in supplemental materials. 

RC18 :: L484-486: Again, not listed in the tables.

Response :: We would be happy to address this comment with additional context.

---

## [Decision Letter · Decision Letter 2]

3 Sep 2024

PONE-D-23-40092R2Predator gaze captures both human and chimpanzee attentionPLOS ONE

Dear Dr. Whitham,

Thank you for submitting your manuscript to PLOS ONE. After careful consideration, we feel that it has merit but does not fully meet PLOS ONE’s publication criteria as it currently stands. Therefore, we invite you to submit a revised version of the manuscript that addresses the points raised during the review process.

We look forward to receiving your revised manuscript.

Kind regards,

Nick Fogt

Academic Editor

PLOS ONE

Journal Requirements:

Additional Editor Comments:

Thank you for your thorough responses to the reviewer comments. Please address the remaining reviewer comments from the latest review.

Reviewers' comments:

Reviewer's Responses to Questions

**Comments to the Author**

1. If the authors have adequately addressed your comments raised in a previous round of review and you feel that this manuscript is now acceptable for publication, you may indicate that here to bypass the “Comments to the Author” section, enter your conflict of interest statement in the “Confidential to Editor” section, and submit your "Accept" recommendation.

Reviewer #2: All comments have been addressed

2. Is the manuscript technically sound, and do the data support the conclusions?

Reviewer #2: Yes

3. Has the statistical analysis been performed appropriately and rigorously? 

Reviewer #2: Yes

4. Have the authors made all data underlying the findings in their manuscript fully available?

Reviewer #2: Yes

5. Is the manuscript presented in an intelligible fashion and written in standard English?

Reviewer #2: Yes

6. Review Comments to the Author

Reviewer #2: Dear authors,

Thank you for addressing the comments and suggestions provided in the previous round of review. I have reviewed the revised version of your manuscript and am pleased to note the improvements made.

The manuscript is now significantly clearer and more transparent, making it much easier to read and understand. The adjustments you have made have strengthened the overall quality of the paper. I appreciate the effort you have put into refining the content, particularly in the areas of data presentation, separating the two experiments and interpretation of the results.

Overall, I am satisfied with the revisions and believe the manuscript is now in a much better state for publication. I have no further major concerns, and I commend you on the work done to enhance the clarity and rigor of the research presented.

Minor comments:

L113-115: Redundant? Has been discussed earlier. Maybe can be integrated.

L327: I assume you used corrected p-values? I would suggest reported what significance values were handled to enhance transparency to the readers. Perhaps I missed it.

L354: I don’t know if the journal follows APA-style, but there the z value is reported before the p value.

L495: I know that the human participants rarely gazed at the backgrounds, but for consistency I think it'd be better to provide this information for Exp1 as well.

L575: I would be careful calling the designs similar.

7. PLOS authors have the option to publish the peer review history of their article (what does this mean?). If published, this will include your full peer review and any attached files.

Reviewer #2: No

---

## [Author Response · Author response to Decision Letter 2]

16 Sep 2024

Reviewer #2: Dear authors,

RC1: Thank you for addressing the comments and suggestions provided in the previous round of review. I have reviewed the revised version of your manuscript and am pleased to note the improvements made. The manuscript is now significantly clearer and more transparent, making it much easier to read and understand. The adjustments you have made have strengthened the overall quality of the paper. I appreciate the effort you have put into refining the content, particularly in the areas of data presentation, separating the two experiments and interpretation of the results. Overall, I am satisfied with the revisions and believe the manuscript is now in a much better state for publication. I have no further major concerns, and I commend you on the work done to enhance the clarity and rigor of the research presented.

Response: We thank the reviewer for their constructive comments throughout

Minor comments:

L113-115: Redundant? Has been discussed earlier. Maybe can be integrated.

Response: We have moved these citations about human and other primates’ apparent bias to faces a few lines higher (ll 109-112) to sit after “These findings suggest that, as with human faces, the bias to attend to nonhuman animal faces is especially strong when direction of gaze is aimed toward the observer.”

L327: I assume you used corrected p-values? I would suggest reported what significance values were handled to enhance transparency to the readers. Perhaps I missed it.

Response: Bonferroni-correct p values were used for all inferences over the post hoc contrasts (ll 365, 545)

L354: I don’t know if the journal follows APA-style, but there the z value is reported before the p value.

Response: We have rearranged these reports to position z values before p (ll 357-359, ll 537-538).

L495: I know that the human participants rarely gazed at the backgrounds, but for consistency I think it'd be better to provide this information for Exp1 as well.

Response: We have added specific means, standard deviations, and confidence intervals to the Experiment 1 Animals vs Backgrounds sections as well (ll 319-321).

L575: I would be careful calling the designs similar.

Response: We have removed this language.

---

## [Editor Report · Decision Letter 3]

24 Sep 2024

Predator gaze captures both human and chimpanzee attention

PONE-D-23-40092R3

Dear Dr. Whitham,

We’re pleased to inform you that your manuscript has been judged scientifically suitable for publication and will be formally accepted for publication once it meets all outstanding technical requirements.

Kind regards,

Nick Fogt

Academic Editor

PLOS ONE

Additional Editor Comments (optional):

Thank you for your responses to the latest reviewer comments.
---

## [Editor Report · Acceptance letter]

8 Oct 2024

PONE-D-23-40092R3 

PLOS ONE

Dear Dr. Whitham, 

I'm pleased to inform you that your manuscript has been deemed suitable for publication in PLOS ONE. Congratulations! Your manuscript is now being handed over to our production team.

Kind regards, 

on behalf of

Dr. Nick Fogt 

Academic Editor

PLOS ONE